# Decomposed Prompt Decision Transformer for Efficient Unseen Task Generalization

**Hongling Zheng**[1]    **Li Shen**[2†]    **Yong Luo**[1†]    **Tongliang Liu**[3]    **Jialie Shen**[4]    **Dacheng Tao**[5]

[1]Wuhan University    [2]Shenzhen Campus of Sun Yat-sen University    [3]The University of Sydney
[4]City, University of London    [5]Nanyang Technological University
{hlzheng, luoyong}@whu.edu.cn    {mathshenli, jialie, dacheng.tao}@gmail.com
tongliang.liu@sydney.edu.au

## Abstract

Multi-task offline reinforcement learning aims to develop a unified policy for diverse tasks without requiring real-time interaction with the environment. Recent work explores sequence modeling, leveraging the scalability of the transformer architecture as a foundation for multi-task learning. Given the variations in task content and complexity, formulating policies becomes a challenging endeavor, requiring careful parameter sharing and adept management of conflicting gradients to extract rich cross-task knowledge from multiple tasks and transfer it to unseen tasks. In this paper, we propose the Decomposed Prompt Decision Transformer (DPDT) that adopts a two-stage paradigm to efficiently learn prompts for unseen tasks in a parameter-efficient manner. We incorporate parameters from pre-trained language models (PLMs) to initialize DPDT, thereby providing rich prior knowledge encoded in language models. During the **decomposed prompt tuning phase**, we learn both cross-task and task-specific prompts on training tasks to achieve prompt decomposition. In the **test time adaptation phase**, the cross-task prompt, serving as a good initialization, were further optimized on unseen tasks through test time adaptation, enhancing the model's performance on these tasks. Empirical evaluation on a series of Meta-RL benchmarks demonstrates the superiority of our approach. The project is available at `https://github.com/ruthless-man/DPDT`.

## 1  Introduction

The purpose of offline reinforcement learning (Offline RL) [1] is to develop a reward-maximizing RL strategy using offline data. This approach is of highly valuable in real-world scenarios where online data collection is expensive, time-consuming, or impractical. Existing offline RL algorithms typically perform well in single task but often struggle for multiple tasks with similar conditions and objectives, as they lack the ability to separate common knowledge from conflicting task gradients. In contrast, humans can leverage knowledge from existing tasks to excel in new ones, which has led to increasing research interest in multi-task reinforcement learning (MTRL) [2, 3, 4]. The goal of MTRL is to develop a universal strategy applicable to tasks with certain similarities, thereby enhancing adaptability and performance in multi-task environments.

Decision transformer (DT) [5] and Prompt-DT [6] introduce the transformer architecture to the field of RL, demonstrating the powerful data modeling capability of sequence offline RL. Additionally, they provide possibilities for integrating advancements [7] from language modeling into MTRL methodologies. Prompt-tuning DT [8] uses a gradient-free method to introduce prompts, retaining context-specific information and catering to specific preferences. These existing sequence-based offline RL methods are primarily trained and tested on the same task or perform fine-tuning using a

---

[†]Corresponding authors.

38th Conference on Neural Information Processing Systems (NeurIPS 2024).

small portion of labeled test data [9]. As a result, these algorithms often perform poorly when faced with testing tasks that are unseen and unlabeled in a Meta-RL setting [10].

Leveraging MTRL to extract general knowledge offers a promising approach for facilitating cross-task knowledge transfer in Meta-RL scenarios. However, as the number of tasks increases, gradient conflicts become more pronounced, hindering MTRL performance due to unregulated parameter sharing. Additionally, while transformer architectures can capture extensive relationships in offline sequential data, their data-hungry nature means that insufficient training data for RL tasks significantly diminishes model performance. A feasible solution might be to enhance the model's prior knowledge.

To remedy these drawbacks, we propose a prompt-based MTRL method named Decomposed Prompt Decision Transformer (DPDT), inspired by some works in natural language processing where knowledge transfer in multi-task learning scenarios is achieved through prompting strategies [11, 12]. We first employ pre-trained parameters from GPT to initialize a DPDT architecture. Incorporating the parameters of PLMs is motivated by several studies in RL [13, 14, 15]. Leveraging the rich prior knowledge encoded in Pre-trained Language Models (PLMs) effectively addresses the data hunger challenge of transformer architectures, providing ample semantic information for reinforcement learning tasks. Then, we design a two-stage training and testing framework for DPDT. (1) **Decomposed prompt tuning phase:** At this stage, we use prompt decomposition to avoid gradient conflicts between different tasks and to extract common knowledge. Specifically, we decompose the task prompt for each task into a cross-task prompt and a task-specific prompt. The cross-task prompt remains consistent across all training tasks, while the task-specific prompt is tailored to each task's unique characteristics. By isolating the cross-task prompt from the task-specific prompts, the model ensures that updates related to general knowledge do not conflict with those related to specific tasks. Compared to [12], our structured decomposition enables more regulated and harmonious parameter updates, thereby enhancing parameter efficiency and facilitating the extraction of general knowledge more effectively. (2) **Test time adaptation phase:** The cross-task prompt, serving as a strong initialization, is further optimized on unlabeled unseen tasks by incorporating the Test Time Adaptation (TTA) [16] to our model. TTA dynamically adjusts the cross-task prompts during the testing phase based on task characteristics, enhancing the model's adaptability to unseen tasks features.

Our DPDT has been empirically validated in the Meta-RL setting, and the results demonstrate its superiority compared with many recent and competitive counterparts [5, 6, 17]. Furthermore, we conducted ablation experiments covering aspects such as prompt length, scalability, and model variants to establish the superiority of the model. The main contributions of this work are as follows:

- We reconsider the problem of knowledge extraction in MTRL and propose the method of Decomposed Prompt Decision Transformer.
- We propose a two-stage paradigm, which includes decomposing the task prompts for training tasks into cross-task prompts and task-specific prompts, and aligning the cross-task prompts in unlabeled unseen tasks.
- We demonstrate the effectiveness of DPDT through intensive experiments on a broad spectrum of benchmarks, highlighting its competitive performance in Meta-RL scenarios.

## 2 Related works

In this section, we summarize the most related works as two-fold: offline RL and multitask RL.

**Offline RL.** In contrast to traditional RL methods [18, 19], offline RL focuses on training models and performing trial-and-error using offline data without environmental interaction to arrive at appropriate strategies. These methods primarily address the issue of out-of-distribution (OOD) through strategies such as constraining the learning policies [20] or bounding the overestimated policy values [21]. The integration of transformer architecture in sequence modeling has emerged as a prominent approach for addressing offline RL tasks [22, 23], further demonstrating the advantages of data-driven policy learning. Decision Transformer (DT) [5] involves encapsulating rewards, states, and actions into triples and training them using autoregressive supervision on offline data. Owing to the transformer's proficiency in capturing and fitting long time series features, it has achieved remarkable results in various offline RL tasks. HDT [17] generalizes new tasks by designing an adaptation module initialized by a hypernetwork. To alleviate the tuning burden while preserving performance, prompt tuning [24, 25] in NLP focuses on optimizing only the input parameters while keeping the majority

of the PLMs parameters frozen. However, integrating prompt tuning into RL field poses a challenge, as RL prompts lack semantic information and are challenging to optimize. While Prompt-DT [6] selects pre-defined expert trajectories combined with inputs to guide model training, it primarily relies on the quality of these trajectories for improvement, rather than enhancing prompts directly. On the other hand, Prompt-tuning DT [8] stands out for introducing prompt-tuning techniques using a gradient-free approach, with the goal of preserving environment-specific details and accommodating specific preferences.

**Multitask RL.** Multi-Task Reinforcement Learning (MTRL) [26, 27, 28] aims to address multiple similar reinforcement learning tasks using a unified model. A straightforward approach involves developing a task-conditional multi-task model, akin to those utilized in goal-conditional RL [29] and visual-language grounding [30]. While this method has demonstrated success in certain scenarios, it often encounters challenges stemming from negative interference among tasks. PaCo [31] delves into a compositional structure within the parameter space, distinguishing between task-agnostic and task-specific components. This approach significantly enhances the efficiency and robustness of the MTRL process, leading to more effective training outcomes. Building upon the MTRL paradigm, multi-task prompt [32] aims to acquire transferable, cross-task prompts from multiple tasks, guiding outputs for unseen downstream tasks. Several studies have approached multi-task prompt design from the perspective of prompt decomposition, yielding notable results across various tasks [33, 34].

The most relevant work to ours is Prompt-DT [6], which utilizes carefully selected prompts for Meta-RL tasks training. Our approach differs in (1) employing trainable prompt decomposition to avoid gradient conflicts among multiple tasks, (2) further optimizing general prompts with TTA without using any test data labels, and (3) extending the model architecture by incorporating PLMs for initialization. To the best of our knowledge, we are the first to implement multi-task prompt tuning based on PLMs parameters in the reinforcement learning domain.

## 3 Preliminary

In this section, we provide several concepts and terminologies that will be used in this work.

### 3.1 Prompt Decision Transformer

Prompt-DT [6] examines the beneficial effects of incorporating trajectory prompts on the DT [5] in few-shot scenarios. Specifically, the form of trajectory prompts is the same as that of training trajectories, consisting of triplets made up of state $s^*$, action $a^*$, and return-to-go $\hat{r}^*$. However, these trajectory prompts are significantly shorter than the training trajectories. During training, for each task $i$, the trajectory prompt $\tau_{i,K^*}^*$ is concatenated with the corresponding training trajectory $\tau_{i,K}$ to form $\tau_i^{input} = \left(\tau_{i,K^*}^*, \tau_{i,K}\right)$, which is then inputted into the model $f$ for training. The concrete form of $\tau_{i,K^*}^*$ and $\tau_{i,K}$ are as follows:

$$\tau_{i,K^*}^\star = (\hat{r}_1^\star, s_1^\star, a_1^\star, \ldots, \hat{r}_{K^\star}^\star, s_{K^\star}^\star, a_{K^\star}^\star), \quad \tau_{i,K} = (\hat{r}_1, s_1, a_1, \ldots, \hat{r}_K, s_K, a_K) \tag{1}$$

where $K^*$ represents the number of environment steps stored in the prompt, and $K$ is the nearest steps of the training trajectory. The prediction head associated with the state token $s$ is designed to predict the corresponding action $a$. It is noteworthy that during training, the model does not predict the actions of the trajectory prompts. The loss function is formulated as follows: $a_{i,m}$ denotes the actual action at the $m$-th timestep of the $i$-th task, while $\tau_{i,m-1}$ encompasses all data up to and including the $(m-1)^{th}$ timestep in the training trajectory of the $i^{th}$ task.

$$L_m = \mathbb{E}_{\tau_i^{input} \sim \mathcal{T}_i} \left[ \frac{1}{K} \sum_{m=1}^{K} \left(a_{i,m} - f(\tau_{i,K^\star}^\star, \tau_{i,m-1})\right)^2 \right] \tag{2}$$

### 3.2 Test-Time Adaptation

Test time adaptation (TTA) [16] aims to minimize the gap between training data and testing data distribution during the testing phase. Test-time prompt tuning (TPT) [35] leverages the extensive knowledge in transformer architecture to enhance its generalization capabilities in zero-shot scenarios.

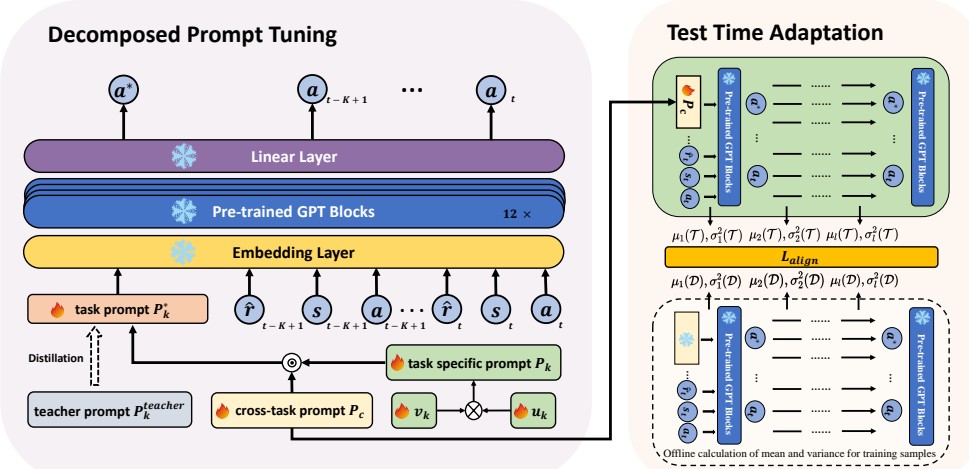

Figure 1: The architecture diagram of DPDT. For simplicity and clarity, the figure only displays the decomposition and integration process during training on a single task. Snowflake icons represent the frozen parts of the model that are not subject to training updates, while flame icons indicate components of the model that remain trainable. **Left: Decomposed Prompt Tuning.** The prompt decomposition is trained on the entire dataset with the assistance of the teacher prompt $p_k^{teacher}$. $p_k^*$ is then combined with the training samples that include $K$ steps, inputted into DPDT, and outputs the corresponding action $a$. **Right: Test Time Adaptation.** When test samples are fed into the model, we calculate the mean and variance of all samples at each layer and compute the loss by comparing them with the mean and variance of the corresponding layer from the training samples. The losses from all layers are summed to obtain the alignment loss, denoted as $L_{align}$.

During the inference phase, multiple randomly augmented views are created from the provided testing sample $X_{test}$. Predictions with entropy below a specified threshold are retained, while other views are filtered out using a confidence selection criterion. The average entropy of the filtered predictions is then used to unsupervisedly update the prompts $p$. Some methods [36, 37, 38], considering the data instability caused by augmentation techniques, use alignment of feature values in attention layers or feedforward layers to achieve test-time adaptation. Our method falls into this category.

## 4 Decomposed Prompt Decision Transformer

In this section, we provide a comprehensive description of the proposed DPDT. Task prompts and training data are combined to calculate the loss through action prediction, which is then used to optimize the cross-task and task-specific prompts in a backward pass. Once optimized, the cross-task prompts serve as a good initialization for use in unseen tasks during test-time adaptation. The objective of decomposed prompt tuning is to learn cross-task prompts and task-specific prompts through prompt decomposition. Test-time adaptation provides an alignment approach for the cross-task prompts. Below, we will describe each module of DPDT in detail.

### 4.1 Decomposed Prompt Tuning

**Initialization.** Given that Transformers are data-intensive and require pre-training on substantial datasets to achieve satisfactory performance, integrating PLMs from the same architectural family into offline RL is a natural progression. Some existing work has already explored this avenue [13, 39, 40]. Taking inspiration from this, we employ GPT2-SMALL [41] to initialize our DPDT and maintain these parameters frozen throughout training. It is worth noting that, incorporating PLMs into RL is not predicated on the direct applicability of language data to RL tasks. Instead, the advantage lies in leveraging the deep, nuanced representations acquired by PLMs from a variety of datasets. These representations encode a broad spectrum of patterns, relationships, and contexts that can transcend purely linguistic tasks. This taps into the reasoning and few-shot capabilities of language models, addressing challenging scenarios like data scarcity and sparse rewards.

**Prompt Decomposition.** Given a set of training tasks $S = \{S_1, S_2, \ldots, S_n\}$, our objective is to learn a general prompt $P_c$ that encapsulates common knowledge shared across all tasks in $S$ and can efficiently adapt to unseen tasks. Extracting general task information from tasks with different distributions is often challenging, as gradient conflicts between tasks can lead to suboptimal convergence of information. We adopt prompt decomposition approach to address this issue. As shown in Figure 1, let $P_c \in \mathbb{R}^{l \times s}$ and $P_k \in \mathbb{R}^{l \times s}$. The task prompt $P_k^*$ for the $k$-th task is obtained by taking the element-wise product of $P_c$ and $P_k$. The goal of prompt decomposition is to enable efficient knowledge sharing across $S$, while still allowing each task to maintain its own parameters to encode task-specific knowledge. The $P_c$ aims to acquire general knowledge from $S$, while the task-specific prompt $P_k$ allows task $k$ to retain its unique knowledge. The task prompts parameterization of the $k$-th training task is expressed as:

$$P_k^* = P_c \circ P_k = P_c \circ (v_k \otimes u_k). \tag{3}$$

In the specific implementation process, inspired by LORA [42], we further decompose each task-specific prompt into two low-rank vectors $v_k \in \mathbb{R}^{l \times r}$ and $u_k \in \mathbb{R}^{l \times s}$ using a low-rank method. We obtain $P_k$ through vector multiplication. Here $l$ represents the prompt length, $r$ represents the hidden layer dimension, and $s$ represents the prompt dimension. The hyperparameter $r$ is a manually specified low-rank parameter. Its introduction is crucial for designing prompts

---

**Algorithm 1** Decomposed Prompt Tuning
___
**Input**: Training task set $S$, Offline datasets $\mathcal{D}_{\mathcal{M}}$, Batch size $M$, Learning rate $\alpha$, training iterations $N$, teacher task prompts $p_k^{teacher}$.
**Initialize**: Initialize a 12-layer, 12-head DPDT $\mathcal{M}$ using GPT2-SMALL, randomly initialize cross-task prompts $P_c$ and low-rank vectors $v_k, u_k$.
    **for** $t = 1$ to $N$ **do**
        **for** $k$ in $S$ **do**
            Select a trajectory $\tau$ that contains $M$ samples in task $k$.
            Calculate $P_k^*$ by Equation 3.
            Calculate $L_{MSE}$ and $L_{dis}$ according to Equations 4 and 5.
            Computed loss function by Equation 6.
            $\theta \leftarrow \theta - \alpha \nabla_\theta \mathcal{L}_{Total}$.
        **end for**
    **end for**

---

for all tasks in the dataset, significantly maintaining model superiority while reducing computational load. We use standard normal distribution to initialize $P_c$, $u_k$ and $v_k$. Given the DPDT $\mathcal{M}$, the mean squared loss between the model's predicted actions and the true actions is calculated as:

$$\mathcal{L}_{MSE} = (a - \mathcal{M}(P_k^\star, \tau))^2 \tag{4}$$

**Prompt distillation.** Due to the lack of explicit constraints, directly implementing prompt decomposition on the multitask dataset $S$ may lead to an overlap in the information learned by $P_c$ and $P_k$, potentially undermining their ability to capture distinct intended details. We employed knowledge distillation techniques to compel the cross-task and task-specific prompts to learn their respective information. We obtained teacher task prompts $p_k^{teacher}$ for each task $k$ by using traditional prompt-tuning methods individually. During training, the mean squared error is calculated directly between $p_k^{teacher}$ and $p_k^\star$:

$$\mathcal{L}_{dis} = \sum_{k \in |\mathcal{S}|} |p_k^{teacher} - p_k^\star|^2 \tag{5}$$

The total loss function for training task prompts for obtaining a cross-task prompt to be transferred to the target side is then:

$$\mathcal{L}_{Total} = \mathcal{L}_{MSE} + \lambda \mathcal{L}_{dis} \tag{6}$$

where $\lambda$ is a weight to balance the impact of distillation loss terms. In our experiments, we set $\lambda$ to 0.5. The overall summary of the multitask training algorithm is presented in Algorithm 1.

## 4.2 Test Time Adaptation

During the test time adaptation (TTA) phase, we address distribution bias by aligning the distribution of unlabeled test samples with the training samples. For each test task $t$ in the test task set $T$, we randomly select a subset $X$ of unlabeled test samples, combine them with the cross-task prompts $P_c$ and input them into the model.

Here, we introduce the data collection method for $X$. The model's testing phase usually occurs in a simulated environment where we predefine our expected reward values $\hat{r}$. The environment

provides the initial state $s$ of the environment, consistent with the settings during inference in prompt DT methods. However, unlike in training tasks where ground-truth labels exist, for action $a_1$, we assign a value sampled randomly from the action space (which is typically consistent between training and testing tasks). We feed this sequence of Markov chains into the environment, obtaining rewards and the next environment states iteratively, assigning a randomly sampled value to action $a_2$ in subsequent iterations. This process is repeated $|X|$ times, resulting in data of the form $(\hat{r}0, s_0, a_0, \hat{r}1, s_1, a_1, \ldots, \hat{r}_{|N|}, s_{|N|}, a_{|N|})$.

In each layer of the model, we calculate the alignment loss based on the means and variances of the training and test samples. Our goal is to update the $P_c$ for the given test task through this alignment loss. For each test task $t$, we denote the distribution of the test samples as $\mathcal{T}$ and the distribution of the training samples as $\mathcal{D}$. Specifically, we calculate the aligned token mean and variance via:

$$\mu_l(\mathcal{T}) = \frac{1}{|X|} \sum_{i=1}^{|X|} H_{l,i}, \quad \sigma_l^2(\mathcal{T}) = \frac{1}{|X|} \sum_{i=1}^{|X|} [H_{l,i} - \mu_l(\mathcal{T})]^2 \tag{7}$$

Here, $H_{l,i}$ represents the state of the $i^{th}$ sample at the $l^{th}$ hidden layer, while $\mu_l(\mathcal{T})$ and $\sigma_l^2(\mathcal{T})$ denote the mean and variance of all test samples at the $l^{th}$ hidden layer, respectively. Similarly, for each hidden layer of the model, we pre-calculate the statistical measures of mean $\mu_l(\mathcal{D})$ and variance $\sigma_l^2(\mathcal{D})$ of the training samples, which are uniformly sampled across all tasks in the training set, in an offline setting to reduce parallel computing costs, since both training samples and labels are accessible. The formula for calculating the alignment loss function is as follows:

---

**Algorithm 2** Test Time Adaptation

**Input**: Test samples set $X$, Cross-task prompts $P_c$, $\mu_l(\mathcal{D})$, $\sigma_l^2(\mathcal{D})$, The number of layers $L$.
1: **for** $l = 1$ to $L$ **do**
2:     **for** $i$ in $X$ **do**
3:         Calculate $H_{l,i}$ obtained by inputting the concatenation of $P_c$ and $i$ into DPDT.
4:     **end for**
5: **end for**
6: **for** $l = 1$ to $L$ **do**
7:     Compute $\mu_l(\mathcal{T})$ and $\sigma_l^2(\mathcal{T})$ by Equation 7.
8: **end for**
9: Compute token distribution alignment loss by Equation 8.
10: Optimize $L_{\text{align}}$ to update $P_c$.

---

$$L_{\text{align}} = \frac{1}{L} \sum_{l=1}^{L} \left( \|\mu_l(\mathcal{T}) - \mu_l(\mathcal{D})\|_1 + \|\sigma_l^2(\mathcal{T}) - \sigma_l^2(\mathcal{D})\|_1 \right). \tag{8}$$

The test time adaptation phase process is illustrated in Algorithm 2.

## 5 Experiment

In this section, we present an extensive evaluation of our proposed DPDT using widely recognized benchmarks. Additionally, we conduct empirical ablation studies to dissect and understand the individual contributions of the core components of our methodology.

### 5.1 Environments and Baselines

**Environments.** To ensure a fair comparison with existing multi-task offline reinforcement learning algorithms, we conducted verification of DPDT using the MuJoCo [43] and MetaWorld [30] benchmarks, which serve as standard tasks in the domain of sequence offline RL, offering sufficient diversity and representing common challenges in classical RL, such as sparse rewards, complex state spaces, and precise control of robotic systems. Our experiments on the Cheetah-dir, Cheetah-vel, and Ant-dir environments in the MuJoCo benchmark meticulously adhere to the datasets and methodologies outlined in Prompt-DT. These tasks penalize agents for using excessive control signals. In the MetaWorld benchmark, we used the ML10, ML45, MT10 and MT50 environments for Meta-RL. A detailed description of the datasets and the division of training and test tasks is provided in Appendix A.

**Baselines.** We compared DPDT to the following offline RL baselines: (1) **Multi-task Behaviour Cloning (MT-BC) [44]**: MT-BC optimizes multi-task learning by exclusively simulating trajectories from the original dataset, dispensing with the need for prompts and reward-to-go tokens. This approach emphasizes the utilization of intrinsic task-specific information, adopting a behavior cloning

Table 1: Results for Meta-RL control tasks (zero-shot scenarios). The best mean accumulated returns are highlighted in bold. For each prompt-needing environment, prompts of length K=30 are utilized. Each experiment was run three times to ensure stability and reproducibility of the results. We report the average returns and standard deviations for these three runs (the higher, the better).

| | MT-BC [44] | MT-ORL [5] | Soft-Prompt [45] | HDT [17] | Prompt-DT [6] | DPDT-WP | DPDT |
|---|---|---|---|---|---|---|---|
| Trainable Params | 125.5M | 125.5M | 3.94M | 12.94M | 125.5M | 1.42M | 1.42M |
| Percentage | 100% | 100% | 3% | 10.31% | 100% | 1.14% | 1.14% |
| **Cheetah-dir** | $-24.71_{\pm12.04}$ | $-86.92_{\pm15.51}$ | $-4.21_{\pm5.51}$ | $-45.32_{\pm13.22}$ | $-7.92_{\pm2.97}$ | $11.73_{\pm12.8}$ | $\mathbf{50.32_{\pm11.47}}$ |
| **Cheetah-vel** | $-201.66_{\pm30.27}$ | $-148.24_{\pm22.18}$ | $-171.23_{\pm20.58}$ | $-162.75_{\pm20.50}$ | $-192.38_{\pm11.80}$ | $-143.14_{\pm21.40}$ | $\mathbf{-139.88_{\pm19.65}}$ |
| **Ant-dir** | $131.89_{\pm12.96}$ | $109.21_{\pm9.66}$ | $119.45_{\pm14.2}$ | $115.43_{\pm10.22}$ | $\mathbf{123.46_{\pm10.70}}$ | $101.49_{\pm17.74}$ | $121.84_{\pm8.01}$ |
| **MW ML10** | $256.77_{\pm11.93}$ | $343.16_{\pm9.40}$ | $246.42_{\pm24.60}$ | $292.14_{\pm8.21}$ | $317.31_{\pm14.98}$ | $204.88_{\pm28.96}$ | $\mathbf{371.01_{\pm9.41}}$ |
| **MW ML45** | $287.37_{\pm11.38}$ | $266.74_{\pm25.81}$ | $91.97_{\pm14.11}$ | $274.88_{\pm19.74}$ | $294.55_{\pm8.71}$ | $300.71_{\pm15.74}$ | $\mathbf{347.21_{\pm11.52}}$ |
| **MW MT 10** | $547.83_{\pm11.04}$ | $1064.58_{\pm21.70}$ | $201.23_{\pm7.11}$ | $964.57_{\pm15.34}$ | $1087.54_{\pm17.09}$ | $1015.91_{\pm0.74}$ | $\mathbf{1317.52_{\pm8.22}}$ |
| **MW MT 50** | $582.80_{\pm13.48}$ | $929.74_{\pm22.81}$ | $400.71_{\pm26.40}$ | $820.45_{\pm27.19}$ | $994.63_{\pm5.99}$ | $1131.01_{\pm1.17}$ | $\mathbf{1559.94_{\pm2.49}}$ |
| **Average** | 225.76 | 354.04 | 130.62 | 309.79 | 373.88 | 374.66 | **518.28** |

strategy to streamline the learning process. (2) **Multi-Task Decision Transformer (MT-ORL) [5]**: We train a decision transformer to learn multiple tasks from the training set. To construct the MT-DT, we exclude prompt augmentation present in DPDT, while retaining the rest of the training process identical to that of DPDT. (3) **Soft-Prompt [45]**: Soft-prompt is trained using a universal prompt across all tasks. (4) **Hyper-decision transformer (HDT) [17]**: HDT efficiently adapts DT to new tasks by augmenting them with an adaptation module, whose parameters are initialized by a hyper-network, enabling quick and efficient adaptation with minimal data. (5) **Prompt-DT [6]**: Prompt-DT builds on DT, leveraging trajectory prompts and reward-to-go for multi-task learning and generalization to unseen tasks.

**Implementation details.** All experiments were carried out on a server with 8 NVIDIA 3090 GPUs, each with 24GB of memory, using PyTorch [46] and Hugging Face Transformers libraries [47]. The experimental hyperparameter configurations are shown in Appendix B. The computer resources utilized by all methods are shown in Table 12.

## 5.2 Main Results and Analysis

**Zero-shot Generalization.** In Table 1, we compare the zero-shot generalization ability of DPDT and the baselines to investigate the overall performance of DPDT. For evaluation, we use the average episode cumulative returns in the test task set as the evaluation metric. Additionally, we introduce a variant of DPDT that does not use GPT-SMALL parameters for initialization, referred to as DPDT-WP (DPDT-Without Pretrained). Soft prompts adapt to tasks in a parameter-efficient way. However, training a universal prompt across multiple tasks suffers from significant gradient interference, as demonstrated by the experimental results. The prompt-DT performs well in few-shot scenarios due to its utilization of test data for fine-tuning. However, in downstream tasks where data is scarce, the prompt fails to provide sufficient task-specific guidance to the model, resulting in suboptimal performance. Importantly, our proposed DPDT exhibits significant performance improvements over fine-tuning and prompt-based methods. This vividly demonstrates the distinct advantages offered by our innovative multitask training techniques. It is worth noting that without initialization with PLM, the performance of DPDT-WP is inferior to most methods. We believe this is mainly due to the model lacking sufficient prior knowledge for effective multi-task prompt tuning, resulting in suboptimal performance of DPDT. In Figure 2, we illustrate the accumulated returns curves of DPDT and other baselines across the Cheetah-vel, MW ML45, and MW MT50 environments. Additional curves for other environments can be found in Figure 4. We also conducted experiments on Soft-Prompt and Prompt-DT combined with TTA (shown in Figure 11). Soft-Prompt-TTA showed performance improvements across all tasks, whereas Prompt-DT-TTA experienced performance declines in some tasks. The main reason for this is that Prompt-DT relies on high-quality trajectory data for prompts during testing, and applying TTA on unlabeled data may have adversely affected prompt optimization.

**Few-shot Generalization.** We explored the performance of DPDT in few-shot scenarios and further investigated whether the prompt decomposition mechanism successfully isolated general knowledge. In this scenario, DPDT does not use TTA to align cross-task prompts $P_c$. Instead, $P_c$ is fine-tuned

Table 2: Results for Meta-RL control tasks (few-shot scenarios). The best mean accumulated returns are highlighted in bold. For each prompt-needing environment, prompts of length K=30 are utilized. Each experiment was run three times to ensure stability and reproducibility of the results. We report the average returns and standard deviations for these three runs (the higher, the better).

| | MT-ORL [5] | Soft-Prompt [45] | HDT [17] | Prompt-DT [6] | DPDT-WP | DPDT | DPDT-F |
|---|---|---|---|---|---|---|---|
| Trainable Params | 125.5M | 3.94 M | 12.94 M | 125.5M | 1.42M | 1.42M | 125.5M |
| Percentage | 100% | 3% | 10.31% | 100% | 1.14% | 1.14% | 100% |
| **Cheetah-dir** | $-46.22_{\pm3.44}$ | $940.24_{\pm1.08}$ | $875.23_{\pm4.24}$ | $934.78_{\pm5.33}$ | $946.81_{\pm17.24}$ | $\mathbf{955.17_{\pm8.03}}$ | $1037.85_{\pm5.98}$ |
| **Cheetah-vel** | $-146.64_{\pm2.12}$ | $-41.81_{\pm2.10}$ | $-63.81_{\pm6.30}$ | $-37.80_{\pm2.09}$ | $-48.07_{\pm1.85}$ | $\mathbf{-30.73_{\pm1.88}}$ | $-29.85_{\pm9.46}$ |
| **Ant-dir** | $110.51_{\pm2.2}$ | $379.01_{\pm1.75}$ | $361.49_{\pm5.63}$ | $\mathbf{411.96_{\pm9.28}}$ | $308.10_{\pm5.22}$ | $384.29_{\pm10.91}$ | $400.01_{\pm9.79}$ |
| **MW ML10** | $421.22_{\pm9.21}$ | $379.82_{\pm14.76}$ | $467.81_{\pm3.07}$ | $315.07_{\pm6.17}$ | $485.27_{\pm19.31}$ | $\mathbf{535.52_{\pm17.39}}$ | $670.24_{\pm3.88}$ |
| **MW ML45** | $264.14_{\pm9.67}$ | $448.72_{\pm11.38}$ | $477.19_{\pm2.16}$ | $473.34_{\pm4.12}$ | $519.28_{\pm7.22}$ | $\mathbf{579.09_{\pm10.42}}$ | $600.44_{\pm17.48}$ |
| **Average** | 120.60 | 421.204 | 423.56 | 419.47 | 442.27 | **484.66** | 535.74 |

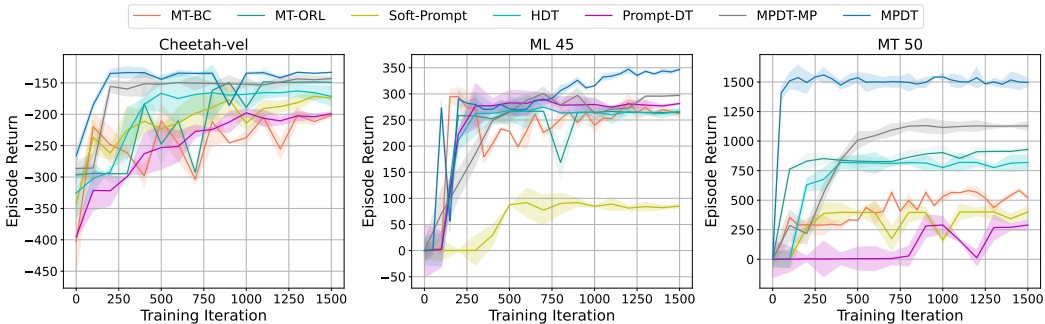

Figure 2: Episodic accumulated returns in three tasks of MTBC, MT-ORL, Soft-Prompt, HDT, Prompt-DT, DPDT-WP and DPDT. Each method is restricted to 1500 rounds of runs in each environment.

directly on a small number of labeled test samples through a self-supervised paradigm. Specifically, we randomly selected only one trajectory from the test dataset for fine-tuning. Other methods adhered to the same amount of fine-tuning data. Furthermore, in few-shot scenarios, we fully fine-tuned DPDT on the complete training and testing data, denoted as DPDT-F. The performance of the DPDT-F method represents the upper bound of all model performances in the current environment. Table 2 shows that the DPDT method, even after fine-tuning, still significantly outperforms or matches the baseline algorithms, demonstrating the effectiveness of prompt decomposition in few-shot environments. Moreover, in some datasets, DPDT approaches the performance of fully fine-tuned models on the test set using only 1.14% of the parameters, as observed in the cheetah-vel environment. It is worth noting that in the Ant-dir environment, the performance of DPDT is slightly inferior to that of Prompt-DT in both zero-shot and few-shot settings. We believe the primary reason for this is the significant domain difference between the language model and environment.

## 5.3 Further Analysis

In the ablation studies, we conducted research on the components of DPDT as well as the impact of prompt length and model parameters on convergence speed and performance.

**Impact of model components.** As shown in Table 3, the impact of prompt decomposition was evaluated. Compared to the soft-prompt method in the unseen task settings (first row), substituting it with decomposed prompts $P_k$ and $P_c$ without distillation (third row) resulted in performance improvements across all three tasks. This ablation highlights the significance of the prompt decomposition strategy in DPDT, demonstrating that the shared component adeptly captures the diverse cross-task knowledge essential for enhancing target downstream tasks.

To evaluate the impact of prompt distillation, we trained a standard prompt shared across all training tasks using the same training loss as DPDT. The teacher prompts for each task remained consistent in DPDT. Compared to the basic baseline (first row), incorporating prompt distillation (second row)

Table 3: Ablation: The impact of prompt decomposition, prompt distillation and test time adaptation.

| Decomposition | Distillation | TTA | Cheetah-vel | MW ML45 | MW MT50 |
|:---:|:---:|:---:|:---:|:---:|:---:|
| ✗ | ✗ | ✗ | -171.23 | 91.97 | 400.71 |
| ✗ | ✔ | ✔ | -163.05 | 108.01 | 709.81 |
| ✔ | ✗ | ✔ | -160.10 | 273.99 | 1137.39 |
| ✔ | ✔ | ✗ | -167.80 | 149.21 | 824.07 |
| ✔ | ✔ | ✔ | **-139.88** | **347.21** | **1559.94** |

Table 4: Ablation: The impact of model size. The elements of the triplet represent, in order, the number of transformer blocks, the count of attention heads, and the size of the hidden layers.

| Model size | Cheetah-vel | Ant-dir | MW ML45 | MW MT50 |
|:---:|:---:|:---:|:---:|:---:|
| (3,1,128) | -164.88 | 129.34 | 288.14 | 749.18 |
| (12,12,768) | **-139.88** | 121.84 | **347.21** | **1559.94** |
| (24,16,768) | -210.35 | **165.99** | 292.48 | 1527.34 |

resulted in a modest improvement in average performance across the three tasks. This emphasizes that prompt distillation from separately trained source prompts is an effective strategy for acquiring high-quality decomposable prompts.

To compare the impact of using TTA on model performance, we examined the results in the fourth and fifth rows. We found that the use of TTA affects the model's final performance. The reason is intuitive: cross-task prompts provide a good initialization environment for TTA, but relying solely on the general information from cross-task prompts is insufficient for the model to perform well on unseen tasks. Incorporating TTA allows the model to adapt to the specific nuances of each task during testing, resulting in substantial performance gains.

**Impact of prompt length.** We examined the influence of prompt length on the performance of DPDT by investigating five distinct prompt lengths (3, 6, 30, 60, 90). Specifically, we explored the effect of prompt length variation on the convergence behavior and generalization capability of the model. It is widely recognized that prompt lengths that are excessively short may impede model convergence, while prompt lengths that are overly long can result in slow convergence rates and potential overfitting. Ablation experiments revealed that a prompt length of 30 is optimal. Further increasing the prompt length to 60 or 90, however, results in minor performance fluctuations but increases the convergence time. Therefore, we used a prompt length of 30 for all our experiments.

**Impact of model size.** We explored the performance of DPDT under three model size configurations. The (3,1,128) configuration uses the pretrained model provided in the original Prompt-DT [6] to initialize DPDT, while the (24,16,768) configuration employs GPT-MIDDLE. Table 4 shows that the size of the pretrained model parameters is correlated with the performance improvement of DPDT. When the model size is expanded to a certain extent (12,12,768), efficient parameter fine-tuning can extract adequate prior knowledge for downstream tasks. However, as the model complexity increases, such as when reaching the size of GPT-MIDDLE, the model exhibits performance improvement on some tasks (Ant-dir) but a decrease in performance on others. This phenomenon can be attributed to the significant gap between the size of the dataset and the complexity of the model. Fine-tuning the model in such scenarios may encounter challenges in appropriately converging for reinforcement learning tasks, potentially leading to overfitting.

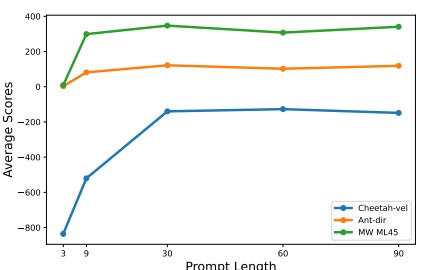

Figure 3: Ablation: The effect of prompt length on DPDT's zero-shot generalization ability.

**Impact of data quality.** The quality of data used for fine-tuning cross prompts does indeed affect the final model performance. We conducted experiments focusing on data quality. In the Cheetah-vel and ML45 environments, we differentiated the quality of datasets into expert, medium, random, and mixed datasets. Each dataset consists of 200 time steps, which aligns with the setup for few-shot scenarios relative to the size of the training set. As shown in Table 5, we found that models fine-tuned

using expert datasets perform the best, which aligns with our intuition. Additionally, the performance of models fine-tuned on mixed datasets is close to that on expert datasets, suggesting implicitly that the DPDT method can extract information from suboptimal datasets to ensure model performance.

**Impact of adaptation method.** In addition to only utilizing cross-task prompts $P_c$ for TTA in zero-shot scenarios, we also investigated (1) combining cross-task prompts $P_c$ with the average of all task-specific prompts $P_k$ from the training set for TTA, (2) freezing the cross-task prompts $P_c$, we initialized a new task-specific prompt combined with the cross-task prompts for TTA and (3) randomly selecting one $P_k$

Table 5: Ablation: The impact of data quality.

|  | Cheetah-vel | ML45 |
| --- | --- | --- |
| expert datasets | -30.10 | 586.84 |
| medium datasets | -41.73 | 502.64 |
| random datasets | -935.66 | 37.91 |
| mixed datasets | -30.73 | 579.09 |

from a training task and combining it with $P_c$ for TTA. However, we found that all of these initialization methods resulted in suboptimal outcomes, shown in Table 10.

**Impact of learning rate in prompt decomposition.** As shown in Table 6, we present experimental results on the ML45 dataset where different learning rates were applied to $P_c$ and $P_k$ in prompt decomposition. We observed optimal performance when both had the same learning rate. We speculate that this occurs because, over training iterations, both prompts converge

Table 6: Ablation: The impact of learning rate in prompt decomposition.

|  | $lr_{P_c}$=1e-2 | $lr_{P_c}$=1e-3 | $lr_{P_c}$=1e-4 |
| --- | --- | --- | --- |
| $lr_{P_k}$=1e-2 | 310.74 | 307.36 | 311.40 |
| $lr_{P_k}$=1e-3 | 198.17 | 350.99 | 338.21 |
| $lr_{P_k}$=1e-4 | 204.94 | 104.07 | 347.21 |

to their optimal values, and differing learning rates disrupt their joint convergence, leading to poorer performance under similar runtime conditions.

**Impact of low-rank parameter $r$.** Table 7 presents the results of our ablation experiments focusing on the low-rank parameter $r$ of prompt decomposition for Cheetah-vel and MW ML45. DPDT is relatively insensitive to selecting hyperparameters, a potential advantage of our work. As observed, the model's performance varies with different values of $r$. These findings suggest that the performance of DPDT remains stable across different values of $r$. This characteristic can be advantageous,

Table 7: Ablation: The impact of low-rank parameter $r$.

|  | Cheetah-vel | MW ML45 |
| --- | --- | --- |
| r=1 | -139.88 | 347.21 |
| r=4 | -138.08 | 344.10 |
| r=10 | -135.51 | 350.33 |

as it allows users to implement the model without extensive tuning, streamlining the deployment process while still achieving competitive results across diverse tasks.

# 6   Conclusion, Limitation and Broader Impact

We have introduced a novel approach, the Decomposed Prompt Decision Transformer (DPDT), aimed at efficient generalization to unseen tasks. Through the utilization of PLMs for parameter initialization and the implementation of parameter-efficient multi-task prompt tuning techniques, we have successfully extracted cross-task general knowledge and further fine-tuned it on previously unseen tasks. Our experiments across various Meta-RL environments demonstrated the effectiveness of our components, achieving superior performance with significantly fewer task-specific parameters compared to fully fine-tuned methods. This approach offers a robust framework for future research to further explore and optimize multi-task learning and generalization capabilities.

**Limitation.** Currently, our work primarily involves using large language models to initialize DPDT. While we've utilized parameter-efficient techniques to fine-tune the model and mitigate inter-domain variances, focusing on optimizing these differences could potentially enhance model performance.

**Broader Impact.** Overall, the application of PEFT methods based on PLMs can help users obtain high-quality reinforcement learning decision models at minimal cost. However, this approach may lead to the misuse of language models when users are unaware of the inter-domain differences, potentially resulting in unforeseen negative outcomes in the RL decision-making process.

# Acknowledgements

This work is supported by the STI 2030-Major Projects (No. 2021ZD0201405), the National Natural Science Foundation of China (Grant No. U23A20318 and 62276195), the Fundamental

Research Funds for the Central Universities (No. 2042024kf0039), the Science and Technology Major Project of Hubei Province under Grant 2024BAB046, and the Innovative Research Group Project of Hubei Province under Grant 2024AFA017. Tongliang Liu is partially supported by the following Australian Research Council projects: FT220100318, DP220102121, LP220100527, LP220200949, IC190100031. Dr. Tao's research is partially supported by NTU RSR and Start Up Grants. The numerical calculations in this paper have been done on the supercomputing system in the Supercomputing Center of Wuhan University.

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

# Appendix

The appendix is organized into several sections, each providing additional insights and details related to different aspects of the main work.

## A  Detailed Environment

- **Cheetah-dir**: There are two tasks in Cheetah-dir with goal directions as forward and backward, respectively. The cheetah agent is rewarded with high velocity along the goal direction. The training and testing set are equal, and both contain the two tasks.

- **Cheetah-vel**: There are 40 tasks in Cheetah-vel with different goal velocities. The target velocities are uniformly sampled from the interval [0,3]. The agent is penalized with l2 errors to the target velocity. We hold out 5 tasks to construct the testing set and train with the remaining 35 tasks.

- **Ant-dir**: There are 50 tasks in Ant-dir with different goal directions uniformly sampled in 2D space. The 8-joints ant is rewarded with high velocity along the goal direction. We sample 5 tasks for testing and leave the rest for training.

- **Meta-World ML10**: In Meta-World ML10, the task is to control a Sawyer robot's end-effector to reach a target position in 3D space. The agent directly controls the XYZ location of the end-effector. Each task has a different goal position. We train in 10 tasks and test in unseen 3 tasks.

- **Meta-World ML45**: In Meta-World ML45, each task has a different goal position. We train in 45 tasks and test in unseen 5 tasks.

- **Meta-World MT10**: Meta-World MT10 comprises 10 distinct robot manipulation tasks with shared dynamics for training and 3 unseen robot manipulation tasks for testing. These tasks exhibit greater variability compared to standard meta-learning environments, posing greater challenges for extracting generalizable knowledge.

- **Meta-World MT50**: The task design of Meta-World MT50 is similar to MT10, with the key difference being an expansion of the training tasks to 45, while the number of unseen test tasks remains at 5.

Adhering to the experimental setup of Prompt-DT, we illustrate the task distribution for both training and testing sets in each dataset, as detailed in Table 13. Our experiments are meticulously crafted in accordance with the specifications outlined in this table.

# B Hyperparameters configuration

We show the hyperparameter of DPDT and other baslines in Table 8 and Table 9.

Table 8: Common Hyperparameters configuration of DPDT and DPDT-WP.

| Hyperparameters | Value |
|---|---|
| Pretraining model | GPT2-small |
| $K$ (length of context) | 20 |
| Prompt Length | 30 |
| training batch size for each task | 16 |
| number of evaluation episodes for each task | 5 |
| learning rate | 1e-4 |
| learning rate decay weight | 1e-4 |
| number of layers | 12 |
| number of attention heads | 12 |
| embedding dimension | 768 |
| activation | ReLU |
| $r$ | 1 |

Table 9: Common Hyperparameters configuration of MT-BC,MT-DT, Soft-prompt, HDT and Prompt-DT.

| Hyperparameters | Value |
|---|---|
| $K$ (length of context $P$) | 20 |
| training batch size for each task | 16 |
| number of evaluation episodes for each task | 5 |
| learning rate | 1e-4 (2e-5 for Prompt-DT) |
| learning rate decay weight | 1e-4 (1e-5 for Prompt-DT) |
| number of layers | 12 |
| number of attention heads | 12 |
| embedding dimension | 768 |
| activation | ReLU |

# C Supplementary experiment

Table 10: Ablation: The impact of adaptation method. (1) Combining cross-task prompts $P_c$ with the average of all task-specific prompts $P_k$ from the training set for TTA, (2) freezing the cross-task prompts $P_c$, we initialized a new task-specific prompt combined with the cross-task prompts for TTA and (3) randomly selecting one $P_k$ from a training task and combining it with $P_c$ for TTA.

| Method | Cheetah-vel | MW ML45 | MW MT50 |
|---|---|---|---|
| (1) | -148.50 | 332.80 | 1482.75 |
| (2) | -171.75 | 320.30 | 1418.50 |
| (3) | -159.26 | 325.02 | 1079.82 |
| DPDT | **-139.88** | **347.21** | **1559.94** |

Table 11: Results for Meta-RL control tasks (zero-shot scenarios).

| | Soft-Prompt-TTA [45] | Prompt-DT-TTA [6] | DPDT |
|---|---|---|---|
| **Cheetah-dir** | $2.91_{\pm 0.74}$ | $8.33_{\pm 0.41}$ | $50.32_{\pm 11.47}$ |
| **Cheetah-vel** | $-160.51_{\pm 10.13}$ | $-204.57_{\pm 8.36}$ | $-139.88_{\pm 19.65}$ |
| **Ant-dir** | $119.14_{\pm 10.72}$ | $120.07_{\pm 2.67}$ | $121.84_{\pm 8.01}$ |
| **MW ML10** | $251.37_{\pm 6.17}$ | $316.74_{\pm 12.05}$ | $371.01_{\pm 9.41}$ |
| **MW ML45** | $301.74_{\pm 18.55}$ | $299.47_{\pm 8.97}$ | $347.21_{\pm 11.52}$ |
| **MW MT 10** | $541.99_{\pm 10.46}$ | $1027.80_{15.58}$ | $1317.52_{\pm 8.22}$ |
| **MW MT 50** | $519.78_{\pm 20.97}$ | $1134.72_{\pm 7.92}$ | $1559.94_{\pm 2.49}$ |
| **Average** | 226.35 | 386.08 | 518.28 |

Table 12: Computer resources (memory, time of execution)

| Method | Batch Size (each) | GPU usage | Wall Time |
|---|---|---|---|
| MT-BC | 16 | 10.4GB | $\approx$ 5-6 hours |
| MT-ORL | 16 | 10.4GB | $\approx$ 1 day |
| Soft-Prompt | 16 | 20.1GB | $\approx$ 5 hours |
| HDT | 16 | 40.8GB | $\approx$ 2 hours |
| Prompt-DT | 16 | 10.9GB | $\approx$ 4 hours |
| DPDT-WP | 16 | 20.5GB | $\approx$ 2-3 hours |
| DPDT | 16 | 20.5GB | $\approx$ 2-3 hours |

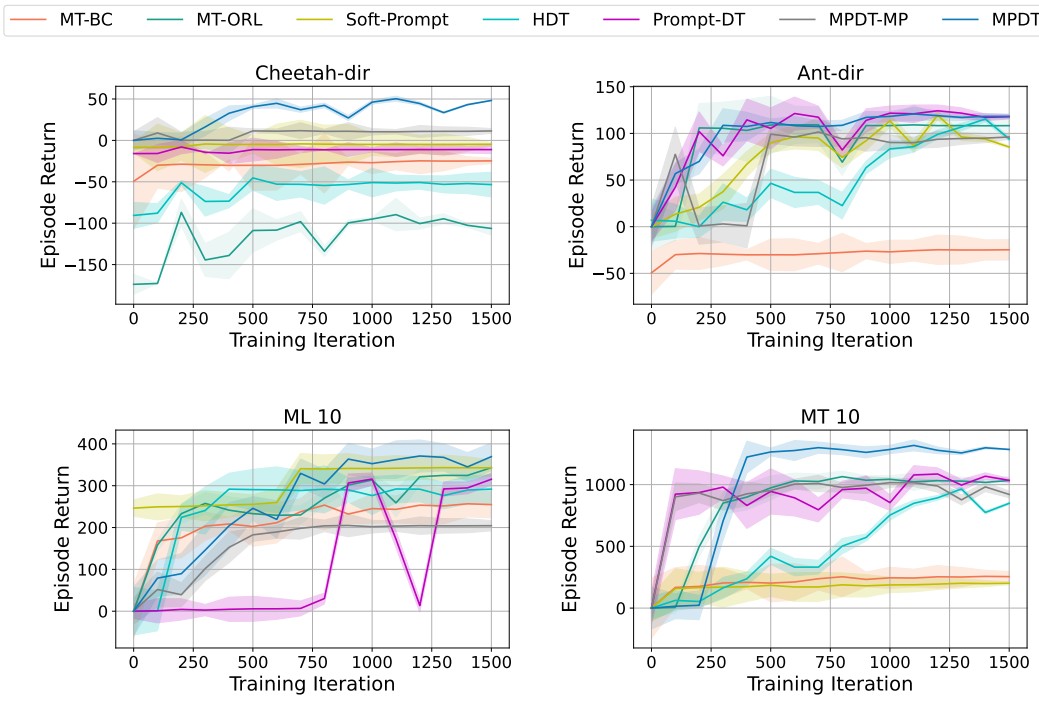

Figure 4: Episodic accumulated returns in four unseen tasks of MTBC, MT-ORL, Soft-Prompt, HDT, Prompt-DT, DPDT-WP and DPDT. Each method is restricted to 1500 rounds of runs in each environment.

Table 13: Training and testing task indexes when testing the generalization ability in unseen tasks.

| Cheetah-dir | |
| --- | --- |
| Training set of size 2 | $[0, 1]$ |
| Testing set of size 2 | $[0.1]$ |

| Cheetah-vel | |
| --- | --- |
| Training set of size 35 | $[0 - 1, 3 - 6, 8 - 14, 16 - 22, 24 - 25, 27 - 39]$ |
| Testing set of size 5 | $[2, 7, 15, 23, 26]$ |

| Ant-dir | |
| --- | --- |
| Training set of size 45 | $[0 - 5, 7 - 16, 18 - 22, 24 - 29, 31 - 40, 42 - 49]$ |
| Testing set of size 5 | $[6, 17, 23, 30, 41]$ |

| Meta-World ML10 | |
| --- | --- |
| Training set of size 10 | $[0, 9, 19, 29, 33, 36, 39, 40, 48, 49]$ |
| Testing set of size 3 | $[11, 24, 41]$ |

| Meta-World ML45 | |
| --- | --- |
| Training set of size 45 | $[0 - 10, 12 - 16, 18 - 24, 26 - 35, 37 - 40, 42 - 49]$ |
| Testing set of size 5 | $[11, 17, 25, 36, 41]$ |

| Meta-World MT10 | |
| --- | --- |
| Training set of size 10 | ["assembly-v2", "button-press-topdown-v2", "coffee-push-v2", "dial-turn-v2", "disassemble-v2", "door-open-v2", "hand-insert-v2", "drawer-open-v2","box-close-v2","push-wall-v2"] |
| Testing set of size 3 | ["peg-unplug-side-v2","hammer-v2","handle-press-v2"] |

| Meta-World MT50 | |
| --- | --- |
| Training set of size 45 | ["basketball-v2", "bin-picking-v2", "button-press-topdown-v2", "button-press-v2", "coffee-button-v2", "coffee-pull-v2", "coffee-push-v2", "dial-turn-v2", "disassemble-v2", "door-close-v2", "door-lock-v2", "door-open-v2", "hand-insert-v2", "drawer-close-v2", "drawer-open-v2", "faucet-close-v2", "handle-press-v2", "handle-pull-side-v2", "handle-pull-v2", "lever-pull-v2","peg-insert-side-v2", "pick-place-wall-v2", "pick-out-of-hole-v2", "reach-v2", "push-back-v2", "push-v2", "pick-place-v2", "plate-slide-v2", "plate-slide-back-v2", "plate-slide-back-side-v2", "soccer-v2", "push-wall-v2", "shelf-place-v2", "sweep-into-v2", "sweep-v2", "window-open-v2", "window-close-v2","assembly-v2","button-press-topdown-wall-v2", "hammer-v2","peg-unplug-side-v2", "reach-wall-v2", "stick-push-v2", "stick-pull-v2", "box-close-v2"] |
| Testing set of size 5 | ["plate-slide-side-v2", "handle-press-side-v2", "buttonpress-wall-v2", "door-unlock-v2", "faucet-open-v2"] |

