# OpenReview forum: "Decomposed Prompt Decision Transformer for Efficient Unseen Task Generalization"
_NeurIPS.cc/2024/Conference — NeurIPS 2024 poster_

### Official Review · Reviewer_d5sD · 2024-06-15

**Soundness:** 3
**Presentation:** 3
**Contribution:** 3
**Rating:** 6
**Confidence:** 4

**Summary:**

This paper proposes the Multi-Task Prompt Decision Transformer (MPDT) algorithm for zero-shot multi-task offline reinforcement learning (RL). Leveraging a pre-trained language model (PLM) with prompt tuning, the MPDT innovatively decomposes multi-task prompts into task-specific and cross-task components. It also achieves zero-shot generalization through test time prompt alignment. Evaluations across various benchmarks demonstrate that MPDT outperforms prior multi-task (meta) offline RL methods.

**Strengths:**

- The paper is well-written and easy to follow.
- The idea of decomposing prompts is straightforward for MTL and easy to understand.
- The experiments are comprehensive.

**Weaknesses:**

- My main concerns come from the novelty of the paper. The major contribution lies in the use of cross-task prompts and task-specific prompts, specifically the prompt decomposition and prompt distillation in Section 4.1.
- Regarding prompt decomposition, although the authors aim for cross-task prompts to contain common knowledge across tasks, the model and loss design do not ensure this. Specifically, since the authors use element-wise multiplication on $P_c$ and $P_k$, $P_c$ functions more as a common scaling factor for different task-specific prompts. After training, $P_c$ could become a constant scalar, and $P_k$ could simply scale as $1/N$ of $P_k^{teacher}$. The authors could provide the distribution of $P_c$ values to verify if it truly encapsulates common knowledge. And it is better to have a loss function to guide the common knowledge extraction during training.
- About the test time adaptation, if the authors believe that $P_c$ contains the common knowledge, I suppose we should train a randomly initialized $P_k$ (or select one from a training task according to task similarity) for the test task using the alignment loss. This way, the model will still take $P_k \cdot P_c$ as prompt input.

**Questions:**

Please refer to the weakness part.

**Limitations:**

The limitations and social impact are provided in the conclusion.

---

> ### Author Rebuttal · Authors · 2024-08-06
>
> Thank you very much for your careful review of our work. We'll answer your questions one by one in the following, including some misunderstandings and some essential academic questions worth exploring.
>
> **W1: About the novelty of the paper**
>
> Please check Author Rebuttal AR1.
>
> **W2: About the prompt decomposition**
> We visualized the cross-task prompt matrix $P_c$ and the task-specific prompts $p_k$ for 10 tasks on MT10, as shown in Figures 1 and 2 in the PDF of Author Rebuttal. From the visual results, $P_c$ exhibits an irregular data distribution, suggesting weighted representations of common knowledge across tasks. In contrast, we found that some $p_k$ matrices for specific tasks show sparsity (values approaching 0), which may explain the essence of prompt decomposition: during training, $P_c$ extracts general features that are then integrated into task prompts $P_k^*$ through element-wise multiplication with sparse $p_k$.
>
> To demonstrate that $P_c$ is more than just a generic scaling factor, we conducted experiments on three datasets: Cheetah-vel, MW ML45, and MT50. We performed fine-tuning without prompt distillation, directly applying $P_c$ element-wise multiplied with the frozen teacher prompts $P_k^*$ for each task. The TTA process remained consistent across experiments. The results are shown in the table below:
>
> |Method|Cheetah-vel|MW ML45|MW MT50|
> |-|-|-|-|
> |$P_c \circ P_k^*$| -169.50|92.88|421.34|
> |MPDT | -139.88 |347.21|1559.94|
>
> If $P_c$ in MPDT is merely a generic scaling factor, we would expect the results in the first row to be similar to MPDT's performance. However, MPDT demonstrates a significant performance advantage, indicating that the prompt decomposition process effectively separates knowledge, aligning with our expectations.
>
> We fully agree that an effective loss function can guide the prompt decomposition process and help the model converge faster. An intuitive idea is to consider adding a sparse loss concerning $p_k$. We incorporate the sum of absolute values of elements in $P_k$ as a penalty term into the loss function, with a weight $\lambda_1$ of 0.2, termed as $L_{sparse}$.
>
> $ L_{sparse}=\lambda_1  \sum_{k=1}^{|S|} \sum_{i=1}^m \sum_{j=1}^n |P_{k_{ij}}|$
>
> We validated the model's performance using $ L_{sparse}$ on the Cheetah-vel and MW ML45 datasets.
> |Method|Cheetah-vel|MW ML45|
> |-|-|-|
> |MPDT+$L_{sparse}$| -138.24.|350.49|
> |MPDT | -139.88 |347.21|
>
> We found that MPDT+$L_{sparse}$ slightly outperforms the original MPDT in terms of performance, implying that the sparsity level of $P_k$ is correlated with prompt decomposition and the model's final performance. Due to rebuttal time constraints, exploration of more instructive loss function designs will be part of our future work. Besides loss design, in our experiments, we tried different learning rates for $P_c$ and $P_k$ but observed optimal performance when both had the same learning rate. We speculate that this occurs because, over training iterations, both prompts converge to their optimal values, and differing learning rates disrupt their joint convergence, leading to poorer performance under similar runtime conditions. Below, we present experimental results on the ML45 dataset where different learning rates were applied to $P_c$ and $P_k$.
>
> | | $lr_{P_c}$=0.01 |$lr_{P_c}$=0.001 |$lr_{P_c}$=0.0001 |
> |-|-|--|--|
> |  $lr_{P_k}$=0.01| 310.74 |307.36 | 311.40|
> | $lr_{P_k}$=0.001| 198.17 | 350.99 | 338.21|
> | $lr_{P_k}$=0.0001| 204.94| 104.07 | 347.21 |
>
> **W3: About the test time adaptation**
> We have already validated similar ideas in the 'Impact of adaptation method' (Section 5.3). We investigated (1) combining $P_c$ with the average of all task-specific prompts $P_k$ from the training set (we call $\hat{P_k}$) for TTA and (2) freezing the $P_c$, randomly initializing a new task-specific prompt $P_r$ combined with $P_c$ for TTA. Following the reviewer's suggestion, we added (3) selecting one $P_k$ from a training task combined with the $P_c$ for TTA, where we randomly sampled three $P_k$ from training tasks and computed the average model performance. We found that both of these initialization methods resulted in suboptimal outcomes. We attribute this primarily to the limited information that TTA can provide [1], where introducing task-agnostic gradients (e.g., additional $p_k$) may significantly degrade model performance. Using $P_c$ for initialization proves to be the optimal approach. Nonetheless, we observed (4) that omitting TTA entirely, i.e., not utilizing any information from test tasks, and instead combining $P_c$ with the average of $p_k$ from training tasks directly for model inference, yields better results than using $P_c$ alone.
>
> |Method|Cheetah-vel|MW ML45|MW MT50 | Is Pc frozen |Using TTA|
> |-|-|-|-|-|-|
> |(1)$P_c \circ \hat{P_k}$|-148.50|332.80|1482.75|×| ✓ |
> |(2) $P_c \circ P_r$|-171.75|320.30|1418.50|✓ | ✓ |
> |(3) $P_c \circ P_k$|-159.26 | 325.02| 1079.82 |×| ✓ |
> |(4) $P_c \circ P_k$ | -167.80|149.21|824.07|×|×|
> |MPDT|$\mathbf{-139.88}$ |$\mathbf{347.21}$|$\mathbf{1559.94}$|×|✓ |
>
> [1] https://link.springer.com/article/10.1007/s11263-024-02181-w
>
> In the case of any follow-up questions, we will in turn provide with further clarifications.

---

> > ### Comment · Reviewer_d5sD · 2024-08-09
> > **Response to the author**
> >
> > Thanks for the response. I've raised my rating to 6.

---

> > > ### Author Response · Authors · 2024-08-10
> > > **Response to the Reviewer d5sD**
> > >
> > > We sincerely thank the reviewer for kindly raising the score.

---

### Official Review · Reviewer_6jy3 · 2024-06-25

**Soundness:** 3
**Presentation:** 3
**Contribution:** 3
**Rating:** 7
**Confidence:** 4

**Summary:**

Multi-task learning is a critical pursuit in decision-making, and Decision Transformer (DT) is a popular framework in solving various decision-making problems. The author observe the suboptimal performance of prior work utilizing DT for multi-task learning, and propose a new method, Multi-Task Prompt Decision Transformer (MPDT), to alleviate this issue. MPDT is mainly composed of these components:
- GPT2 pre-trained weights for initialization.
- Prompt decomposition: cross-task prompt + task-specific prompt to prevent gradient conflicts.
- Test time adaptation (TTA): dynamically optimizes the cross-task prompts during testing time.

Experiments are conducted on standard Meta-RL tasks, demonstrating prominent improvements compared to prior baselines. Ablation experiments are also extensively conducted.

**Strengths:**

Originality:
- The MPDT framework combining pre-trained weights, prompt decomposition and TTA together is original.
- The prompt decomposition technique to prevent gradient conflicts is orginal and well-motivated.

Clarity:
- This paper is well-written and well-organized.
- The method is easy to follow.

Significance:
- Though pre-training DT is a popular trend in decision-making, this is the first work to successfully apply it in multi-task learning, to the knowledge of the reviewer.
- The experimental results are prominent, compared with a set of strong baselines.

**Weaknesses:**

*Significant*
- It is unclear to the reviewer that whether it is fair to compare MPDT with baselines like MT-BC, Soft-Prompt, and Prompt DT. since MPDT adopts much test-time information. How can the authors guarantee that they use the same amount of information in the test set for all baselines? If it is truly unfair, it would make the experiments less convincing. Please provide explanations for this.
- Did the authors reproduce the results of baselines by themselves? If so, how did they pick the hyper-parameters? And for few-shot generalization, which appears to be a new benchmark designed by the authors, did the authors extensively tune the hyper-parameters of baselines?

*Major*
- It is always hard to claim the so-called "SOTA", which demands very rigorous statistical analysis. And it is rarely true in nowadays AI community, see https://rl-conference.cc/review_guidelines.html. Running 3 times is acceptable to present the performance, but not enough to support "SOTA". And the variances in Table 1,2 are very large (some [$\mu-\sigma$, $\mu+\sigma$] intervals are overlapped), thus making it hard to establish statistical improvements. The reviewer recommends removing this claim.
- The framework is a bit complicated and engineering. Specifically, the 3 components, namely pre-training+prompt decomposition+TTA, are orthogonal and not well-connected. And thus the technical novelty of the method is a bit lacking. The novelty concentrates on prompt decomposition+distillation, while solely applying them doesn't achieve good performance (Table 3).


*Minor*
- The wording "Unseen Tasks Generalization" and "Few-shot Generalization" in Section 5.2 are misleading, as few-shot generatlization is also for unseen tasks.
- The name of the proposed method, *Multi-Task Prompt DT*, might not be suitable, since it cannot reflect the differences from Prompt DT, which is already a well-developed algorithm in muti-task RL.

**Questions:**

- It looks unnatural to the reviewer that only the weights of prompts are trainable while the embedding layer and GPT blocks are all frozen, as shown in Figure 1. The ablation experiments of [1][2] show that parameter-efficient tuning like LoRA achieves better performance than freezing the whole block. Can the author provide some ablation experiments or reasonable explanations for this?
- Notably, the ablation experiments in Table 3 and performance of MPDT-WP show that, only putting all these 3 components together can achieve best performance. If these components are really orthogonal and helpful, then why simply using one or two of them won't achieve overall improvements? For example, on MW ML45, only using all 3 components can beat Prompt-DT. Could the author share some insights on this?
- How much efforts did the authors spend in tuning the hyper-parameters of MPDT?

[1] Can Wikipedia Help Offline Reinforcement Learning? arXiv preprint arXiv:2201.12122.

[2] Unleashing the Power of Pre-trained Language Models for Offline Reinforcement Learning. arXiv preprint arXiv:2310.20587.

**Limitations:**

The limitations are discussed in the paper.

---

> ### Author Rebuttal · Authors · 2024-08-06
>
> Thanks for the careful review of our work.
>
> **W1: How can the authors guarantee that they use the same amount of information in the test set for all baselines?**
> In methods not involving prompts, we indeed fine-tuned these methods on the test set. To highlight the superiority of MPDT as much as possible, we allowed for some unfairness in comparison. Therefore, we sampled $|X|$ labelled samples for fine-tuning MT-BC, MT-ORL, and HDT methods, making the experiments in Table 1 persuasive for non-prompt methods.
>
> For methods involving prompts (Soft-Prompt and Prompt-DT),to ensure the absolute fairness of our method, we add experiments combining Soft-Prompt and Prompt-DT with TTA. The experimental results, labelled as Soft-Prompt-TTA and Prompt-DT-TTA, are shown in the table below.
> ||Soft-Prompt-TTA|Prompt-DT-TTA|MPDT
> |-|-|-|-|
> |Cheetah-dir|$2.91\pm0.74$|$8.33\pm0.41$|$50.32\pm 11.47$
> |Cheetah-vel|$-160.51\pm10.13$|$-204.57\pm8.36$|$-139.88\pm19.65$
> |Ant-dir|$119.14\pm10.72$|$120.07\pm2.67$|$121.84\pm8.01$
> |MW ML10|$251.37\pm6.17$|$316.74\pm12.05$|$371.01\pm9.41$
> |MW ML45|$301.74\pm18.55$|$299.47\pm8.97$|$347.21\pm11.52$
> |MW MT 10|$541.99\pm10.46$|$1027.80\pm15.58$|$1317.52\pm8.22$
> |MW MT 50|$519.78\pm20.97$|$1134.72\pm7.92$|$1559.94\pm2.49$
> |Average|$226.35$|$386.08$|$518.28$
>
> Soft-Prompt-TTA showed performance improvements across all tasks, whereas Prompt-DT-TTA experienced performance declines in some tasks. The main reason for this is that Prompt-DT relies on high-quality trajectory data for prompts during testing, and applying TTA on unlabeled data may have adversely affected prompt optimization. Our MPDT method continues to demonstrate advantages across the majority of tasks.
>
> **W2: Did the authors reproduce the results of baselines?**
> We have replicated the experimental results of the baselines. In Table 6 of Appendix B, we provide the configuration of hyperparameters used. For Prompt-DT, MT-BC, and MT-ORL, we use the code provided by the Prompt-DT, adjusting most parameters to ensure a fair comparison with MPDT. Soft-Prompt, based on MPDT, training a universal prompt directly on datasets from all tasks. HDT followed all the details from the original paper. To ensure a fair comparison with MPDT, we aligned some general hyperparameters with those used by MPDT (See Author Rebuttal AR4).
>
> For experiments in the few-shot generalization scenario, we did not adjust any hyperparameters and maintained the configuration from Table 6 because the chosen baselines either had architectures designed for few-shot scenarios or transferable insights applicable to our study.
>
> [1] https://arxiv.org/abs/2304.08487
>
> **W3: Removing the claim of "SOTA".**
> Thanks for the suggestions and comments. We will remove the statement regarding SOTA terminology and clarify that our results are competitive.
>
> **W4: The technical novelty and performance problem.**
> About the technical novelty, please check Author Rebuttal AR1. About the performance problem in Table 3, please check Q2.
>
> **W5: The misleading problem of unseen tasks.**
> We are considering changing "Unseen Tasks Generalization" in Section 5.2 to "zero-shot generalization," so that our whole experiments can be referred to as "Unseen Tasks Generalization."
>
> **W6: The name might not be suitable.**
> Our main innovation lies in the implementation of prompt decomposition, distillation and subsequent TTA alignment in prompts. It can be modified to "Decomposed Prompt Decision Transformer Enables Unseen Task Generalization. This title may be more aligned with the current algorithm.
>
> **Q1: It looks unnatural that only the weights of prompts are trainable while the embedding layer and GPT blocks are all frozen.**
> Please check Author Rebuttal AR3.
>
> **Q2: Why simply using one or two of components won't achieve improvements?**
> The experimental results shown in Table 3 (fourth row) are misleading. The cross-task prompt $P_c$ contains rich inter-task information, providing a strong initialization for adapting to test tasks. To achieve optimal performance, fine-tuning task-specific information with TTA is necessary.
>
> $P_c$ performs poorly in directly generalizing to unseen tasks. However, this does not imply that the prompt decomposition component cannot be used independently. To verify this, we performed experiments where $P_c$ was element-wise multiplied with the average of $P_k$ from all training tasks (referred to as $P_{ka}$ ), directly used for test tasks without TTA. As shown in the table below, our performance still surpasses that of the first row. The original intention of the fourth row in Table 3 was to demonstrate the poor performance of using $P_c$ alone. However, we recognize that this should not be construed as an experiment on the prompt decomposition component. Therefore, we propose replacing the fourth row in Table 3 with the results obtained by using $P_c \circ P_{ka}$ as the prompt for test tasks to avoid confusion.
> |Decomposition|Distillation|TTA|Cheetah-vel|MW ML45|MW MT50| remark
> |-|-|--|-|-|-|-
> |$\times$|$\times$|$\times$|-171.23|91.97|400.71|
> |$\checkmark$|$\checkmark$|$\times$|-145.27|337.80|1304.07|$P_c\circ P_{ka}$
> |$\checkmark$|$\checkmark$|$\checkmark$|$\mathbf{-139.88}$|$\mathbf{347.21}$|$\mathbf{1559.94}$|
>
> **Q3: How much efforts did the authors spend in tuning the hyper-parameters?**
> Almost all hyperparameters are referenced from Prompt DT, and we did not excessively tune our hyperparameters, which facilitates a fair comparison with other methods. The primary hyperparameter we focused on tuning is the prompt length $l$. The ablation experiments of the optimal value for $l$ are shown in Figure 3. In the table below, we present the results of ablation experiments on another hyperparameter $r$ for Cheetah-vel and MW ML45. Overall, MPDT is relatively sensitive to the selection of hyperparameters, which is a potential advantage of our work.
> ||Cheetah-vel|MW ML45
> |-|-|-
> |r=1|-139.88|347.21
> |r=4|-138.08|344.10
> |r=10|-135.51|350.33

---

> ### Comment · Reviewer_6jy3 · 2024-08-10
>
> I thank the authors for the detailed responses to my questions. On reading the responses, most of my concerns have resolved.
>
> Two concerns remain:
> - *(minor)* W1. Prompt DT cannot utilize the same amount of test data due to its design. To alleviate this(?), the authors conduct experiments on Prompt-DT+TTA for fair comparison, but TTA isn't suitable for Prompt-DT, making it not fair enough. (Anyway, this is not a big problem.)
> - *(major)* Q1. The question that "why the GPT blocks are all frozen" is not answered yet.
>
> Another concern:
> - *(significant)* The initial hyperparameter of Prompt-DT is #layer=3, #heads=1. The rebuttal states that "To ensure a fair comparison with MPDT, we aligned some general hyperparameters with those used by MPDT (See Author Rebuttal AR4)", and it hence seems that the experiments of Prompt-DT in this paper use #layer=12, #heads=12. If so, that would be unfair, since the other hyperparameters of Prompt-DT must be tuned extensively due to a significant change in the model size.

---

> ### Author Response · Authors · 2024-08-11
> **Response to the Reviewer 6jy3（1）**
>
> We sincerely appreciate the reviewers' thorough examination. Below, we address the concerns one by one.
>
> **W1: Prompt DT cannot utilize the same amount of test data, and the TTA is unsuitable.**
> The Prompt-DT+TTA method selects the prompt $p$ that allows the model to perform optimally on the test task from the high-quality trajectories of the training tasks and then fine-tunes $P$ on the same amount of unlabeled test data as MPDT for TTA. This approach ensures zero-shot generalization (without using high-quality trajectories from the test task as prompts) and effectively combines Prompt-DT with TTA. We believe this is the optimal way to adapt Prompt-DT to zero-shot scenarios. The performance fluctuations of Prompt-DT+TTA across different tasks are mainly because Prompt-DT is not inherently a prompt-tuning method (despite using frozen prompts), as it requires high-quality labeled trajectories as prompts during both testing and training, which is a very strong prior condition. It would be more accurate to say that Prompt-DT might not be suitable for zero-shot generalization scenarios rather than stating it is unsuitable for TTA. In few-shot scenarios, MPDT still outperforms Prompt-DT, demonstrating MPDT's superior performance in existing scenarios and its ability to extend to new scenarios that existing algorithms struggle to handle effectively. From a rigor perspective, Prompt-DT could be removed from Table 1, which also indirectly highlights the limited exploration of prompt-based offline RL methods in zero-shot generalization scenarios.
>
> **Q1: Why the embedding layer and GPT blocks are all frozen.**
> In Author Rebuttal AR3, we conducted an ablation study on whether to freeze the embedding layer. Also, based on the ablation study conclusions from [1][2], we speculate that the embedding layer and GPT blocks are highly correlated. To achieve sufficient performance, one should either (1) freeze both the embedding layer and GPT blocks simultaneously and add external components (e.g., prompt) or (2) train both the embedding layer and GPT blocks together (either full fine-tuning or using LoRA). The choice of approach involves a trade-off between model performance, computational cost, and scenario requirements.
>
> From the perspective of model performance, the primary purpose of using prompts is to preserve the model's inherent prior knowledge as much as possible. Both [2] and we believe that the full model fine-tuning used in [1] may lead to model overfitting, further disrupting the internal knowledge of the model. We considered adapting the method from [2] to our task (training the embedding layer and LoRA), but the results obtained based on the code provided by [2] differed significantly from those in the original paper. Moreover, when we applied [2] to our datasets, ML45 and MT50 (as shown in the table below), the performance on the training set was far inferior to MPDT, and even careful hyperparameter tuning could not alleviate the significant fluctuations in the reward curve. We speculate that language models using the LoRA structure may still face convergence difficulties on RL data. Furthermore, the experiments in [2] primarily focus on single-task scenarios, avoiding the challenges of multi-task scenarios where task gradients may mix with the internal knowledge of the model.
>
> ||ML45 (Training Performance)|MT50 (Training Performance)
> |-|-|-
> |LAMO[2]| $367.563\pm129.37$ |$ 930.84\pm 437.62$
> |MPDT| $ 604.21\pm11.05$ |  $1687.35\pm4.09$
>
> Furthermore, [2] introduced a language prediction auxiliary loss to ensure the model retains its language task memory during training. However, if the introduction of LoRA is entirely suitable and does not disrupt the internal knowledge of the model, there should be no need for an auxiliary loss. In other words, we must fully understand whether full fine-tuning or LoRA fundamentally introduces new knowledge by disrupting existing knowledge (replacement) or guides the model to use existing knowledge to achieve RL tasks (integration). The distinction between these two approaches results in differences in the model's performance ceiling and floor. The extent of adjustments within the model may need to be carefully controlled. Additionally, current work needs to explore convergence and generalization adequately. MPDT combines transferable prompts with the pre-trained model in a more harmonious and adaptable way. Most importantly, freezing the embedding layer and GPT blocks preserves the model's complete knowledge, which we believe is a better choice.

---

> ### Author Response · Authors · 2024-08-11
> **Response to the Reviewer 6jy3（2）**
>
> From the perspective of computational cost, the advantages of MPDT are clear (1.42M vs. 125.5M fine-tune) or (1.42M vs. 3.5M LoRA). Fully fine-tuning a model is undoubtedly costly, and LoRA's 3.5M parameters are based on the adjustable parameter count in a single-task scenario, as reported in [2]. When extending to a multi-task scenario, the parameter count will increase proportionally to the number of tasks.
>
> From the perspective of scenario requirements, using prompts offers a natural advantage for knowledge transfer. Prompts have sufficient capacity to guide pre-trained models to adapt to reinforcement learning tasks with smaller data sizes. Additionally, achieving zero-shot generalization with prompts results in significantly less disruption to the model's knowledge and a lower error tolerance rate than fully fine-tuning the model.
>
> [1] Can Wikipedia Help Offline Reinforcement Learning
> [2] Unleashing the Power of Pre-trained Language Models for Offline Reinforcement Learning
>
> **Significant concern: The other hyperparameters of Prompt-DT must be tuned due to a change in the model size.**
>
> We did indeed use the 12-layer, 12-head version of Prompt-DT. To ensure a fair comparison, we reran the experiments and carefully adjusted some hyperparameters (setting the learning rate to 2e-5 and the learning rate decay weight to 1e-5). Other hyperparameters are independent of the model size. The model performance did not change significantly under the optimal hyperparameter configuration. The table below shows the performance of Prompt-DT and Prompt-DT-TTA across all tasks after adjusting the hyperparameters. We will replace the corresponding results in the main text with these updated results.
>
> Table 1: Zero-shot generalization
> ||Prompt-DT|Prompt-DT-TTA
> |-|-|-|
> |Cheetah-dir|$-7.92\pm2.97$|$9.03\pm2.11$
> |Cheetah-vel|$-192.38\pm11.80$|$-203.07\pm4.01$
> |Ant-dir|$123.46\pm10.70$|$121.64\pm3.83$
> |MW ML10|$317.31\pm14.98$|$314.08\pm12.93$
> |MW ML45|$294.55\pm8.71$|$294.87\pm10.06$
> |MW MT 10|$1087.54\pm17.09$|$1030.85\pm14.77$
> |MW MT 50|$994.63\pm5.99$|$1137.69\pm13.58$
>
> Table 2: Few-shot generalization
> ||Prompt-DT
> |-|-|
> |Cheetah-dir|$934.78\pm5.33$
> |Cheetah-vel|$-37.80\pm2.09$
> |Ant-dir|$411.96\pm9.28$
> |MW ML10|$315.07\pm6.17$
> |MW ML45|$473.34\pm4.12$

---

> > ### Comment · Reviewer_6jy3 · 2024-08-11
> >
> > Thank you for your insights! From the results, the reviewer guesses that LoRA tuning trick might only be superior when learning single task with limited data. Now all my concerns resolved.
> >
> > I will raise my score to 7.

---

> ### Author Response · Authors · 2024-08-11
> **Response to the Reviewer 6jy3**
>
> We agree with the reviewer's point and sincerely thank the reviewer for raising the score.

---

### Official Review · Reviewer_zhrS · 2024-07-11

**Soundness:** 3
**Presentation:** 2
**Contribution:** 3
**Rating:** 4
**Confidence:** 4

**Summary:**

This paper proposes a new method called Multi-Task Prompt Decision Transformer (MPDT) for efficient generalization to unseen tasks in offline reinforcement learning. MPDT involves two stages. First, the multitask training phase: MPDT is initialized with parameters from a pertained LM, which is GPT2. It decomposes the task prompt into a cross-task prompt shared across tasks and task-specific prompts. Second, the test time adaptation: The cross-task prompt is further optimized on unlabeled test tasks using test time adaptation by aligning the distributions of the test samples and training samples. The paper evaluates MPDT on seven meta RL environments from both MuJoCo and MetaWorld, showing its superior performance over baselines in generalizing to unseen tasks.

**Strengths:**

1. Based on the review's knowledge, combining decision transformer and test time adaptation together for efficient multi-task RL solving is novel.

2. Keeping the weights of the pretrained LM frozen is a natural idea to leverages its rich prior knowledge.

3. Extensive experiments on seven Meta-RL environments demonstrate MPDT's effectiveness over DT-based baselines.

4. Ablation studies individually analyze the impact of different components including prompt decomposition, distillation, and test time adaptation.

Overall, MPDT appears to be a promising approach for multi-task offline RL by combining prompt-based techniques with test time adaptation in a novel way.

**Weaknesses:**

1. The paper appears to be hastily written and the presentation is hard to follow. (see question section)

2. The authors claim that a major improvement compared with other PDTs is the decomposition of prompt into the common part and task-specific part. I was expecting to see the common part will be trained across tasks and the task-specific part is within a specific task. However, in line 186, the authors say ‘We use standard normal distribution to initialize Pc, uk and vk.’ Besides, in Eq(4), they are equally optimized for each task. I didn’t see why the common part is the slow weights and the task-specific part is the fast weight, which is claimed by authors in line 167. Possible to explain here?

3. In line 192, the sentence ‘we obtain teacher task prompt $p_k^{teacher}$ for each task by using traditional prompt-tuning method individually. ‘ What method is used? How to learn it?  Is there a separate learning phase for $p_k^{teacher}$? If so, the method requires three training phases instead of two.  Also its quality matters. Can authors explain the details here?

4. In line 201, for test time adaptation, ‘ we randomly select a subset X of unlabeled test samples,….’ Can authors clarify what is this subset? A sequence in the DT is organized  as (r_t, s_t, a_t, r_{t+1}, s_{t+1}, a_{t+1}…). I recommend being specific what you select?

**Questions:**

1. How do tasks for each meta-environment differ? Do they have different reward functions $R(s_t, a_t)$ or transition function $T(s_t, a_t) \to s_{t+1}$. It question is important for the proposed method MPDT. Because I wonder what kind of distribution shift that test-test can handle and what it cannot.

2. How does the quality of the few shot prompt effect the model’s final performance?  Specifically, using expert chunk, random chunks, medium policy chunks, and mixed quality chunks. It would be better to have some analysis here.

Presentation Questions:

3. In algorithm 1, L_{dis} is calculated without being used?

4. In Eq.(4) What is $M((P_k^*, \tau))$? It is used without any explanation? Also, how many input does M take? Why there is a double layer bracket?

5. In Line 176, the authors state that  $P_k\in \mathbf{R}^{l\times s}$ is the vector multiplication of two low rank vectors $v_k\in R^{l \times r}$ and $u_k \in R^{l\times s}$. First low-rank is a matrix property, there is no low-rank vector. Second, putting the wording issue aside, what is the vector multiplication here? If it is the matrix multiplication, which is used in the low-rank decomposition methods, then the shape of the output should be $r\times s$. Better to clarify here.

**Limitations:**

See my above comments for weakness and questions

---

> ### Author Rebuttal · Authors · 2024-08-06
>
> Thanks for reviewing our work attentively. We will answer the reviewer's questions one by one in the following.
>
> **W1: The paper appears to be hastily written and the presentation is hard to follow**
> We will revise all unclear and erroneous statements highlighted by the reviewer further to improve the logical structure and clarity of the paper.
>
> **W2: The problem about prompt decomposition**
> The significance of prompt decomposition lies in splitting the task prompt $P_k^*$ into cross-task prompts $P_c$ and task-specific prompts $P_k$. Both prompts are trained concurrently during the training process, where data from different tasks are sequentially inputted into the model. For the current task $k$, the model computes $P_k^*$ as the element-wise product of $P_k$ and $P_c$, which is then concatenated with the input data and optimized on the current task. Using standard normal distribution to initialize $P_c$, $u_k$ and $v_k$ is a common practice for prompt initialization [1,2].
>
> $P_c$ is termed "slow" because it captures universal knowledge shared among all tasks in the task set $S$. Its learning rate should be slow to prevent overfitting to specific tasks. On the other hand, $P_k$ is termed "fast" because it compensates by quickly adapting to the characteristics of the current task, aiding better model convergence. In our experiments, we attempted different learning rates for $P_c$ and $P_k$ but found that performance was optimal when both had the same learning rate. We speculate that this is because, over training iterations, both prompts converge to their optimal values, and setting different learning rates disrupts their joint convergence, resulting in poorer performance under similar runtime conditions. Below, we present experimental results on the ML45 dataset where different learning rates were applied to $P_c$ and $P_k$.
> ||$lr_{P_c}$=0.01|$lr_{P_c}$=0.001|$lr_{P_c}$=0.0001|
> |-|-|-|-|
> |$lr_{P_k}$=0.01|310.74|307.36|311.40|
> |$lr_{P_k}$=0.001|198.17|350.99|338.21|
> |$lr_{P_k}$=0.0001|204.94|104.07|347.21|
>
> [1] https://arxiv.org/abs/2109.04332
> [2] https://arxiv.org/abs/2210.02390
>
> **W3: How to obtain teacher task prompt?**
> Here exists a separate training process for $P^{teacher}_k$, which has dimensions $\mathbb{R}^{l\times s}$. Prompt tuning is used to learn $P^{teacher}_k$ independently for each task $k$. Since there are no gradient conflicts in training on individual tasks, this process is fast and straightforward. We set the batch size to 256 and train for 100 epochs, typically converging in about 1 hour. There are indeed three training stages, but we consider the training of $P^{teacher}_k$ as a preparatory data phase. Once learned, $P^{teacher}_k$ does not require retraining.
>
> **W4: Can authors clarify what is subset $X$?**
> Please check Author Rebuttal AR2.
>
> **Q1: How do tasks for each meta-environment differ?**
> Each task within the environment has distinct reward functions and state transition functions.
>
> The Cheetah-dir environment is one-dimensional, rewarding agents based on the angular difference between their movement direction and target direction. The Cheetah-vel environment is also one-dimensional, rewarding agents based on the difference between their velocity and a target velocity. The Ant-dir environment is two-dimensional, rewarding agents based on the angular difference between their movement direction and uniformly sampled target directions within 360 degrees.
>
> ML10, ML45, MT10 and MT50 are trained and tested in a three-dimensional physical space. Each task in these datasets is a goal-conditioned environment, where the state space across all tasks shares the same dimensions. The action space remains identical across different tasks, though specific dimensions in the state space represent different semantic meanings. The task divisions are detailed in Table 9 of Appendix C. Due to space limitations, we describe 14 robotic manipulation tasks in Table 1 of the attached PDF for the Author Rebuttal.
>
> **Q2: How does the quality of the few shot prompt effect the model’s performance?**
> Intuitively, the quality of data used for fine-tuning cross prompts does indeed affect the final model performance. We conducted additional experiments focusing on data quality. In the Cheetah-vel and ML45 environments, we differentiated the quality of datasets into expert, medium, random, and mixed datasets. Specifically, we randomly selected labelled test set trajectories and partitioned the first 30% as random, the middle 30% as a medium, and the last 30% as expert datasets. The mixed datasets are an average blend of these three types. Each dataset consists of 200-time steps, which aligns with the setup for few-shot scenarios relative to the size of the training set.
> ||Cheetah-vel|ML45|
> |-|-|-|
> |expert datasets|-30.10|586.84|
> |medium datasets|-41.73|502.64|
> |random datasets|-935.66|37.91|
> |mixed datasets |-30.73|579.09|
>
> We found that models fine-tuned using expert datasets perform the best, which aligns with our intuition. Additionally, the performance of models fine-tuned on mixed datasets is close to that on expert datasets, suggesting implicitly that the MPDT method can extract information from suboptimal datasets to ensure model performance.
>
> **Q3: $L_{dis}$ is calculated without being used?**
> We are sorry for this mistake. The parameter update rule is $\theta \leftarrow \theta-\alpha \nabla_\theta \mathcal{L}_{Total}$.
>
> **Q4: Why there is a double layer bracket?**
> $\mathcal{M}(P_k^*,\tau)$ here denotes the concatenation of task prompt $P_k^*$ and trajectory $\tau$ inputted into MPDT $\mathcal{M}$.
>
> **Q5: What is the vector multiplication here?**
> The dimension of $u_k$ should indeed be $\mathbb{R}^{r\times s}$. Furthermore, we should refer to both $v_k$ and $u_k$ as low-rank matrices rather than vectors, as these parameters exhibit matrix properties. Therefore, $P_k\in\mathbb{R}^{l\times s} $ is obtained through matrix multiplication $P_k = v_k\otimes u_k$.

---

> ### Author Response · Authors · 2024-08-14
> **Respectful Request for Reviewer‘s Valuable Feedback on Our Rebuttal**
>
> Dear Reviewer zhrS,
>
> We sincerely appreciate the time and effort you have already dedicated to reviewing our submission. We have carefully considered and addressed your initial concerns regarding our paper. Given that the discussion phase is nearing its end, we would greatly appreciate it if you could share any further feedback so we can respond promptly. Additionally, we welcome any new suggestions or comments you may have.
>
> Thank you once again for your valuable insights and continued support.
>
> Best regards,
> The authors of Submission 9319

---

### Official Review · Reviewer_urHM · 2024-07-13

**Soundness:** 3
**Presentation:** 2
**Contribution:** 3
**Rating:** 4
**Confidence:** 4

**Summary:**

This paper proposes a novel Multi-Task Prompt Decision Transformer (MPDT), which leverages pre-trained language models as the initialization and adopts test-time adaptation. This approach achieves efficient generalization to unseen tasks through the prior knowledge from the pre-trained language model and by decomposing the task prompt. In the multi-task training stage, the prompt is decomposed into a cross-task prompt and a task-specific prompt, which can reduce gradient conflicts and computational load. Besides, the task-specific prompt is further decomposed into two low-rank vectors. Prompt distillation is also used to improve the quality of the prompt decomposition. In test-time adaptation, the method further optimizes the cross-task prompts on unseen tasks via alignment loss. An empirical study on Meta-RL benchmarks demonstrates the superior performance of MPDT compared to existing methods.

**Strengths:**

- This paper leverages GPT as the initialization to provide the prior knowledge for RL tasks, which is reasonable. The empirical study demonstrates that MPDT with the pre-trained language model initialization outperforms MPDT without the language model initialization.
- This paper utilizes prompt decomposition and only updates the prompt parameter in the multi-task training, which significantly reduces the trainable parameters. Compared with baselines, MPDT with fewer trainable parameters achieves superior performance.
- The empirical study demonstrates that MPDT outperforms the baselines, and further analysis demonstrates the effect of the components in this method.

**Weaknesses:**

- This paper proposed utilizing the pre-trained language model as initialization but lacks an explanation of why the language pre-trained model could contribute to the RL tasks.
- This method needs a long prompt to perform well compared with the Prompt DT.
- This paper lacks detailed explanation on why using the cross-task prompt, task-specific prompt, and the prompt distillation.

**Questions:**

- Can you explain why you use the pre-trained embedding layer for word tokens to encode the RL tokens?
- Can you explain more about why you use prompt decomposition and prompt distillation?
- Can you explain why you need to use a long prompt to achieve better performance?

**Limitations:**

See weaknesses and questions.

---

> ### Author Rebuttal · Authors · 2024-08-06
>
> Thanks for reviewing our work attentively. We'll answer your questions one by one in the following. Considering the word limitation, we combine Weaknesses and questions with similar meanings to answer.
>
> **W1: why the language model could contribute to the RL tasks?**
> In Section 4.1, the Initialization part, we explained the reasons for using language models for initialization，"Given that Transformers are data-intensive and require pre-training on substantial datasets to achieve satisfactory performance, integrating PLMs from the same architectural family into offline RL is a natural progression. Incorporating PLMs into RL is not predicated on the direct applicability of language data to RL tasks. Instead, the advantage lies in leveraging the deep, nuanced representations acquired by PLMs from a variety of datasets. Theserepresentations encode a broad spectrum of patterns, relationships, and contexts that can transcend purely linguistic tasks."
>
> Additionally, incorporating the parameters of pre-trained language models is also motivated by several studies in RL[1,2]. Leveraging the rich prior knowledge encoded in PLMs effectively addresses the data hunger challenge of transformer architectures, providing ample semantic information for RL tasks. These papers demonstrate the advantages of using language models for RL. Our approach differs from previous work by freezing the core architecture of the language model and using trainable prompts to guide the model's knowledge for RL tasks. This approach reduces computational demands (Table 7) while enhancing transferability. Experimental results in Table 1 comparing MT-ORL, MPDT-WP, and MPDT further validate the benefits of initializing MPDT with PLMs.
>
> [1] https://arxiv.org/abs/2310.20587
> [2] https://arxiv.org/abs/2201.12122
>
> **W2 & Q3: Why you need to use a long prompt to achieve better performance?**
> In Prompt DT, prompts used during training and testing consist of carefully selected high-quality trajectory segments. The original paper of Prompt DT does not specify the exact method for selecting high-quality prompts, and since the main focus of training is the model itself rather than the prompt, the requirements for prompt length are relatively low. Furthermore, the performance of Prompt DT requires sufficient support from prior information (labeled test data), as illustrated in Figure 3 of the Prompt DT paper, where prompt quality significantly impacts model performance, especially in scenarios with poor data quality.
>
> The key difference in the MPDT method lies in freezing the core model, significantly reducing parameter count compared to Prompt DT (1.42M vs 125.5M). More than directly comparing prompt lengths between Prompt DT and MPDT may be unfair. On the other hand, MPDT optimizes prompts initialized completely at random on training data, thus requiring almost no prior knowledge.
>
> In Figure 3 (with fixed training duration), we find that a prompt length of around 30 performs best. Without considering computational costs and training time constraints, we reintroduce performances of prompt lengths 3 and 9 on two tasks (trained until model convergence), finding that a prompt length around 9 also achieves highly competitive results. Moreover, prompt lengths between 9 and 30 are feasible in scenarios using a language model, as they do not noticeably increase the computational burden.
>
> | prompt length | Cheetah-vel | ML45 |
> |---------------|-------------|------|
> | 3             | -299.20 (about 8.5 h) | 110.83 (about 11.0 h) |
> | 9             | -141.49 (about 4.8 h) | 348.03 (about 6.7 h) |
> | 30            | **-139.88** (about 2.4 h) | **347.21** (about 4.9 h) |
>
> [1] https://arxiv.org/abs/2206.13499
>
> **W3 & Q2: This paper lacks detailed explanation on why using the cross-task prompt, task-specific prompt, and the prompt distillation.**
> In the context of multitask reinforcement learning, conflicting gradients arising from different tasks due to their objectives and environments are key factors leading to poor model performance. Let us separate common features and conflicting features between tasks. In that case, it not only facilitates training the model on existing tasks but also allows us to use common features (manifested as prompts in our work) for prompt initialization in downstream tasks. The cross-task prompt remains consistent across all training tasks, while the task-specific prompt is tailored to each task's unique characteristics. Our structured decomposition enables more regulated and harmonious parameter updates, thereby enhancing parameter efficiency and facilitating the extraction of general knowledge more effectively.
>
> The primary purpose of prompt distillation is to aid in further separating two types of prompts. Due to the lack of explicit constraints, directly implementing prompt decomposition on the multitask dataset $S$ may lead to an overlap in the information learned by $P_c$ and $P_k$, potentially undermining their ability to capture distinct intended details. We found that employing knowledge distillation from prompts trained separately on individual training tasks was a successful approach for obtaining well-decomposable prompts.
>
> We verify the performance of the proposed component in Table 3 of the main text.
>
> **Q1: Why you use the pre-trained embedding layer for word tokens to encode the RL tokens?**
> Please check Author Rebuttal AR3.

---

> > ### Comment · Reviewer_urHM · 2024-08-13
> >
> > Thank the authors for their detailed responses and additional experiments. The Rebuttal AR3 experiment demonstrates the performance of sticking the word embedding. However, the explanation can not convince me.  I will keep the score.

---

> ### Author Response · Authors · 2024-08-13
> **Response to the Reviewer urHM**
>
> We sincerely appreciate the reviewers' thorough examination and hold the reviewers' opinions in the highest regard. We strive to do our utmost to address reviewers' concerns, not only about the acceptance of the paper but also in providing meaningful insights for the RL community.
>
> The reviewer's concern can be understood as the possibility that not retraining certain layers of the language model may result in an inadequate understanding of RL data by the model. We believe preserving the language model's knowledge is crucial, and prompts can bridge the gap between the two tasks without retraining the embedding layer or GPT blocks. In AR3, we conducted an ablation study on whether to freeze the embedding layer. The experimental results are intuitive, as training only the embedding layer while freezing the GPT blocks still leads to a suboptimal understanding of input information. Ablation study conclusions from [1][2] also demonstrate that the embedding layer and GPT blocks are highly correlated ([1] performs full fine-tuning of a language model on RL tasks. In contrast, [2] trains a language model on RL tasks using LoRA). Comparing [1][2] with MPDT can further address the reviewer's concerns. The reviewer's concerns can be extended to whether to (1) freeze both the embedding layer and GPT blocks simultaneously and add external components (e.g., prompt) or (2) train both the embedding layer and GPT blocks together (either full fine-tuning or using LoRA). Only these two approaches can achieve better performance. The choice of approach involves a trade-off between model performance, computational cost, and scenario requirements. Our decision to adopt approach (1) is based on the following reasons.
>
> From the perspective of model performance, the primary purpose of using prompts is to preserve the model's inherent prior knowledge as much as possible. Both [2] and we believe that the full model fine-tuning used in [1] may lead to model overfitting, further disrupting the internal knowledge of the model. We considered adapting the method from [2] to our task (training the embedding layer and LoRA), but the results obtained based on the code provided by [2] differed significantly from those in the original paper. Moreover, when we applied [2] to our datasets, ML45 and MT50 (as shown in the table below), the performance on the training set was far inferior to MPDT, and even careful hyperparameter tuning could not alleviate the significant fluctuations in the reward curve. We speculate that language models using the LoRA structure may still face convergence difficulties on RL data.
>
> ||ML45 (Training Performance)|MT50 (Training Performance)
> |-|-|-
> |LAMO[2]| $367.563\pm129.37$ |$ 930.84\pm 437.62$
> |MPDT| $ 604.21\pm11.05$ |  $1687.35\pm4.09$
>
> Furthermore, [2] introduced a language prediction auxiliary loss to ensure the model retains its language task memory during training. However, if the introduction of LoRA is entirely suitable and does not disrupt the internal knowledge of the model, there should be no need for an auxiliary loss. In other words, we must fully understand whether retraining the model internally (including the embedding layer or GPT blocks) fundamentally introduces new knowledge by disrupting existing knowledge (replacement) or guides the model to use existing knowledge to achieve RL tasks (integration). The distinction between these two approaches results in differences in the model's performance ceiling and floor. The extent of adjustments within the model may need to be carefully controlled. Most importantly, freezing the embedding layer and GPT blocks preserves the model's complete knowledge, which we believe is a better choice, and the experimental results also support this.
>
> From the perspective of computational cost, the advantages of MPDT are clear (1.42M vs. 125.5M fine-tune) or (1.42M vs. 3.5M LoRA). Fully fine-tuning a model is undoubtedly costly, and LoRA's 3.5M parameters are based on the adjustable parameter count in a single-task scenario, as reported in [2]. When extending to a multi-task scenario, the parameter count will increase proportionally to the number of tasks.
>
> From the perspective of scenario requirements, prompts have sufficient capacity to guide pre-trained models to adapt to reinforcement learning tasks with smaller data sizes. Additionally, achieving zero-shot generalization with prompts results in significantly less disruption to the model's knowledge and a lower error tolerance rate than fully fine-tuning the model.
>
> We demonstrated from both experimental results and theoretical principles whether it is necessary to freeze the embedding layer and GPT blocks. We sincerely welcome further discussion with the reviewer and will try to address any concerns.
>
> [1] Can Wikipedia Help Offline Reinforcement Learning? arXiv preprint arXiv:2201.12122.
> [2] Unleashing the Power of Pre-trained Language Models for Offline Reinforcement Learning. arXiv preprint arXiv:2310.20587.

---

### Author Rebuttal · Authors · 2024-08-06

We thank all the reviewers for their helpful feedback. Here, we address three main comments: innovation (AR1), the collection method of unlabeled data for TTA (AR2), and the purpose of using a pre-trained embedding layer (AR3). AR4 is the list of hyperparameters used during training and testing. The remaining questions and concerns are addressed in individual responses. In the attached PDF, we provide visualizations of the cross-task prompt and the task-specific prompts for MT10, along with introductions to 14 tasks in the MT50 dataset. All of these additional experiments and suggestions have been added to the updated main text.

**AR1:**
The MPDT framework proposed by us encompasses the following innovations:
- Building upon prior knowledge from pre-trained models, we innovatively propose prompt decomposition, accelerating convergence through prompt distillation. Our idea of learning a transferable cross-task prompt by decomposing and distilling knowledge from multitask datasets is unique, which not only makes the prompt learning more performant but results in fewer parameters.

- Current TTA feature alignment techniques primarily focus on directly optimizing model feature layers. In scenarios with extremely limited unlabeled test samples, we innovatively use cross-task prompts as a robust initialization. By utilizing alignment-based reverse optimization of prompts, we preserve the language model's capabilities and condense task-specific information into cross-task prompts. Disruptive experimental results in the table below demonstrate that using randomly initialized prompts for TTA initialization severely compromises model performance.

- Our innovations interlock seamlessly, tightly integrating into an efficient and practical framework, paving the way for future paradigms in multi-task offline reinforcement learning.

|Decomposition| Distillation| TTA|Cheetah-vel| MW ML45 |MW MT50|remark|
|-|-|-|-|-|-|-|
| $\times$ | $\times$ | $\times$|-171.23| 91.97 | 400.71||
| $\times$ |$\times$ |$\checkmark$ | -180.39|99.82 | 644.33 |  Randomly initialized prompt|
| $\checkmark$ | $\checkmark$| $\checkmark$ | $\mathbf{-139.88}$ | $\mathbf{347.21}$ | $\mathbf{1559.94}$ ||

**AR2:**
Here, we introduce the data collection method for $X$. The model's testing phase usually occurs in a simulated environment where we predefine our expected reward values $\hat{r}$. The environment provides the initial state $s$ of the environment, consistent with the settings during inference in prompt DT methods. However, unlike in training tasks where ground-truth labels exist, for action $a_1$, we assign a value sampled randomly from the action space (which is typically consistent between training and testing tasks). Action dimension and action distribution are shown in the table below. We feed this sequence of Markov chains into the environment, obtaining rewards and the next environment states iteratively, assigning a randomly sampled value to action $a_2$ in subsequent iterations. This process is repeated $|X|$ times, resulting in data of the form $(\hat{r}_0, s_0, a_0, \hat{r}_1, s_1, a_1, \ldots, \hat{r}_{|N|}, s_{|N|}, a_{|N|})$. We consider the method of randomly sampling action values similar to assigning pseudo-labels to the data for TTA, helping the prompt understand the characteristics of the current task.

||Action dimension| Action distribution|
|-|-|-|
|Cheetah-dir|6 |[-1,1]|
|Cheetah-vel|6 |[-1,1]|
|Ant-dir|8|[-1,1]|
|MW ML10|4|[-3,3]|
|MW MT45|4|[-3,3]|
|MW MT10|4|[-1,1]|
|MW MT50|4|[-1,1]|

**AR3:** The purpose of using a pre-trained embedding layer is to maximize the retention of the internal information of the language model. Considering the differences between our input $(\hat{r},s,a)$ and natural language inputs, part of the function of the prompt is also to guide the language model in understanding inputs from RL tasks. To validate the effectiveness of using the pre-trained embedding layer, we adopted the RL tokens embedding layer design method from Prompt DT. During multi-task training, we concurrently trained the embedding layer, denoted as MPDT-E. Each experiment was run three times to ensure stability and reproducibility of the results.

|| Cheetah-vel|MW ML45|MW MT50|
|-|-|-|-|
|MPDT-E |-186.07| 301.49| 380.36|
|MPDT|-139.88| 347.21| 1559.94 |

We found that training a new RL tokens embedding layer resulted in performance improvements inferior to using the pre-trained embedding layer. We attribute this mainly to the retraining of the embedding layer, which disrupts the language model's understanding of input knowledge. Additionally, our inputs themselves treat $(\hat{r},s,a)$ as a single token, with temporal relationships between tokens, resembling traditional language input structures to some extent. Simultaneously, guidance from the prompt maximizes bridging the gap between these aspects, preserving the performance gains from the pre-trained embedding layer more effectively than retraining.

**AR4:**
|Hyperparameters|Value
|-|-
|K|20
|demonstration length|20
|training batch size for each task $M$|16
|number of layers|12
|number of attention heads|12
|number of gradient updates in each iteration|5
|number of evaluation episodes for each task|5
|learning rate|1e-4
|learning rate decay weight|1e-4
|activation|RELU

---

### Decision · Program_Chairs · 2024-09-25

**Decision:**

Accept (poster)

**Comment:**

The paper originally received mixed reviews, and concerns regarding:

1. concerns with regard to the comparison to Prompt DT - the main baseline for the paper
2. reliance on long prompts and LLM for initialization
3. clarity
4. fairness of use of test-time adaptation with regards to baselines that do not use it
5. novelty
6. baselines reproduction and choice of HP
7. statistical significance of the gains

the authors seem to have been able to address the concerns of 2/3 reviewers who raised their score to Accept and Weak Accept. However, the reviewer urHM remained unconvinced and provided additional reasons to reject the paper, albeit after the reviewer-authors discussion period concluded, not allowing the authors to respond to these concerns. In light of the above, and observing some overlap between addressed concerns (as confirmed by other reviewers) and new concerns by urHM, the AC is inclined to accept the paper. Yet, the AC urges the authors to seriously consider all concerns raised by the reviewers, and incorporate all the feedback and responses into the revised version of their paper. The AC strongly suggests the authors pay special attention to additional concerns raised by urHM after the discussion period, also addressing them and incorporating them into the final revision.